# FOUNDATION MODEL-ORIENTED ROBUSTNESS: ROBUST IMAGE MODEL EVALUATION WITH PRETRAINED MODELS

Peiyan Zhang[1], Haoyang Liu[2], Chaozhuo Li[3*], Xing Xie[3], Sunghun Kim[1] and Haohan Wang[2*]

[1]Hong Kong University of Science and Technology
[2]University of Illinois at Urbana-Champaign
[3]Microsoft Research Asia

## ABSTRACT

Machine learning has demonstrated remarkable performance over finite datasets, yet whether the scores over the fixed benchmarks can sufficiently indicate the model's performance in the real world is still in discussion. In reality, an ideal robust model will probably behave similarly to the oracle (*e.g.*, the human users), thus a good evaluation protocol is probably to evaluate the models' behaviors in comparison to the oracle. In this paper, we introduce a new robustness measurement that directly measures the image classification model's performance compared with a surrogate oracle (*i.e.*, a zoo of foundation models). Besides, we design a simple method that can accomplish the evaluation beyond the scope of the benchmarks. Our method extends the image datasets with new samples that are sufficiently perturbed to be distinct from the ones in the original sets, but are still bounded within the same image-label structure the original test image represents, constrained by a zoo of foundation models pretrained with a large amount of samples. As a result, our new method will offer us a new way to evaluate the models' robustness performance, free of limitations of fixed benchmarks or constrained perturbations, although scoped by the power of the oracle. In addition to the evaluation results, we also leverage our generated data to understand the behaviors of the model and our new evaluation strategies.

## 1 INTRODUCTION

Machine learning has achieved remarkable performance over various benchmarks. For example, the recent successes of various neural network architectures (He et al., 2016a; Touvron et al., 2021) has shown strong numerical evidence that the prediction accuracy over specific tasks can reach the position of the leaderboard as high as a human, suggesting different application scenarios of these methods. However, these methods deployed in the real world often underdeliver its promises made through the benchmark datasets (Edwards, 2019; D'Amour et al., 2020), usually due to the fact that these benchmark datasets, typically *i.i.d*, cannot sufficiently represent the diversity of the samples a model will encounter after being deployed in practice (Recht et al., 2019; Wu et al., 2023).

Fortunately, multiple lines of study have aimed to embrace this challenge, and most of these works are proposing to further diversify the datasets used at the evaluation time. We notice these works mostly fall into two main categories: (1) the works that study the performance over testing datasets generated by predefined perturbation over the original *i.i.d* datasets, such as adversarial robustness (Szegedy et al., 2013; Goodfellow et al., 2015) or robustness against certain noises (Geirhos et al., 2019; Wang et al., 2020b); and (2) the works that study the performance over testing datasets that are collected anew with a procedure/distribution different from the one for training sets, such as domain adaptation (Ben-David et al., 2007; 2010) and domain generalization (Muandet et al., 2013).

---

*Co-corresponding authors

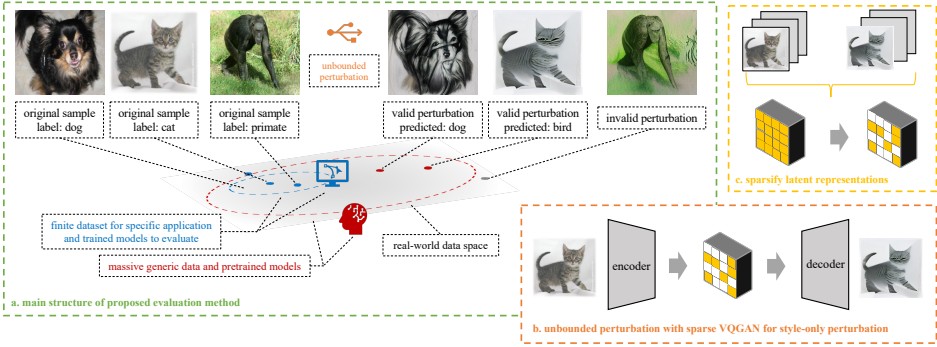

Figure 1: The main structure of our system to generate test images with foundation models and examples of the generated images with their effectiveness in evaluation of model's robustness.

Both of these lines, while pushing the study of robustness evaluation further, mostly have their own advantages and limitations as a tradeoff on how to guarantee the underlying image-label structure of evaluation samples will be the same as the training samples: perturbation based evaluations usually maintain the image-label structure by predefining the perturbations within a set of operations that will not alter the image semantics, such as $\ell$-norm ball constraints (Carlini et al., 2019), or texture (Geirhos et al., 2019), frequency-based (Wang et al., 2020b) perturbations, but are relatively limited in the variety of perturbations allowed. On the other hand, new-dataset based evaluations maintain the image-label structure by soliciting the efforts of human annotators to construct datasets with the same semantics but significantly different styles (Hendrycks et al., 2021b; Hendrycks & Dietterich, 2019). However, such new datasets may be costly to collect, and a potential issue is that they are fixed once collected and published to the research community. Ranking methods based on the fixed datasets will eventually lead to the methods overfit on certain datasetss (Duda et al., 1973; Friedman et al., 2001; Yan et al., 2024). While recent efforts have tried to alleviate the selection bias by collecting data from multiple sources (Gulrajani & Lopez-Paz, 2020; Koh et al., 2021; Ye et al., 2021; Wang et al., 2022b; Li et al., 2019), we kindly argue that a dynamic process of generating evaluation datasets will certainly further mitigate this issue.

In this paper, we investigate how to diversify the robustness evaluation datasets to make the evaluation results credible and representative. As shown in Figure 1, we aim to integrate the advantages of the above two directions by introducing a new protocol to generate evaluation datasets that can automatically perturb the samples to be sufficiently different from existing test samples, while maintaining the underlying unknown image-label structure with respect to a zoo of foundation models. Based on the new evaluation protocol, we introduce a new robustness metric that measures the robustness compared with the foundation model. Moreover, with our proposed evaluation protocol and metric, we make a study of current robust machine learning techniques to identify the robustness gap between existing models and the foundation model. This is particularly important if the goal of a research direction is to produce models that function reliably comparable to the foundation model.

Therefore, our contributions in this paper are three-fold:

- We introduce a new robustness metric that measures the robustness gap between models and the foundation model.
- We introduce a new evaluation protocol to generate evaluation datasets that can automatically perturb the samples to be sufficiently different from existing test samples, while maintaining the underlying unknown image-label structure.
- We leverage our evaluation metric and protocol to conduct the very first systematic study on robustness evaluation. Our analysis brings us the understanding and conjectures of the behavior of the deep learning models, opening up future research directions.

## 2 BACKGROUND

**Current Robustness Evaluation Protocols.** The evaluation of machine learning models in non-*i.i.d* scenario have been studied for more than a decade, and one of the pioneers is probably *domain*

*adaptation* (Ben-David et al., 2010). In domain adaptation, the community trains the model over data from one distribution and tests the model with samples from a different distribution; in *domain generalization* (Muandet et al., 2013), the community trains the model over data from several related distributions and test the model with samples from yet another distribution. To facilitate the development of cross-domain robust image classification, the community has introduced several benchmarks, such as PACS (Li et al., 2017), ImageNet-A (Hendrycks et al., 2021b), ImageNet-C (Hendrycks & Dietterich, 2019), ImageNet-Sketch (Wang et al., 2019), and collective benchmarks integrating multiple datasets such as WILDS (Koh et al., 2021), and OOD Bench (Ye et al., 2021).

While these datasets clearly maintain the underlying image-label structure of the images, a potential issue is that these evaluation datasets are fixed once collected. Thus, if the community relies on these fixed benchmarks repeatedly to rank methods, eventually the selected best method may not be a true reflection of the world, but a model that can fit certain datasets exceptionally well. This phenomenon has been discussed by several textbooks (Duda et al., 1973; Friedman et al., 2001). While recent efforts in evaluating collections of datasets (Gulrajani & Lopez-Paz, 2020; Koh et al., 2021; Ye et al., 2021) might alleviate the above potential hazards of "model selection with test set", a dynamic process of generating evaluation datasets will certainly further mitigate this issue.

On the other hand, one can also test the robustness of models by dynamically perturbing the existing datasets. For example, one can test the model's robustness against rotation (Marcos et al., 2016), texture (Geirhos et al., 2019), frequency-perturbed datasets (Wang et al., 2020b), or adversarial attacks (*e.g.*, $\ell_p$-norm constraint perturbations) (Szegedy et al., 2013). While these tests do not require additionally collected samples, these tests typically limit the perturbations to be relatively well-defined (*e.g.*, a texture-perturbed cat image still depicts a cat because the shape of the cat is preserved during the perturbation).

While this perturbation test strategy leads to datasets dynamically generated along the evaluation, it is usually limited by the variations of the perturbations allowed. For example, one may not be able to use some significant distortion of the images in case the object depicted may be deformed and the underlying image-label structure of the images is distorted. Generally speaking, most of the current perturbation-based test protocols are scoped by the tradeoff that a minor perturbation might not introduce enough variations to the existing datasets, while a significant perturbation will potentially destroy the underlying image-label structures.

**Assumed Desiderata of Robustness Evaluation Protocol.** As a reflection of the previous discussion, we attempt to offer a summary list of three desired properties of the datasets serving as the benchmarks for robustness evaluation:

- **Stableness in Image-label Structure:** the most important property of the evaluation datasets is that the samples must represent the same underlying image-label structure as the training samples.
- **Effectiveness in Generated Samples:** the test samples should be effective in exposing defects for tested models.
- **A Dynamic Generation Process:** to mitigate selection bias of the models over techniques that focus too attentively to the specification of datasets, ideally, the evaluation protocol should consist of a dynamic set of samples, preferably generated with the tested model in consideration.

**Necessity of New Robustness Measurement in Dynamic Evaluation Protocol.** In previous experiments, two settings are commonly used: a "standard" test set and a perturbed test set. Previous approaches rank models based on accuracy under perturbed test set (Geirhos et al., 2019; Hendrycks et al., 2021a; Orhan, 2019; Xie et al., 2020; Zhang, 2019) or other metrics such as inception score (Salimans et al., 2016), effective robustness (Taori et al., 2020) and relative robustness (Taori et al., 2020). While useful for initial assessments, these metrics do not fully capture robustness in dynamic evaluation protocols. Here, comparing two models on different dynamic test sets cannot definitively determine superior model robustness, as differences in performance may result from varying test set difficulties.

The main challenge identified is the lack of a consistent robustness metric across test sets. Ideally, a robust model should mirror the behavior of the foundation model (e.g., human users). Therefore, a direct measurement of model robustness relative to the foundation model is preferable over indirect model comparisons.

## 3 METHOD - COUNTERFACTUAL GENERATION WITH SURROGATE ORACLE

### 3.1 METHOD OVERVIEW

We use $(\mathbf{x}, \mathbf{y})$ to denote an image sample and its corresponding label, and use $\theta(\mathbf{x})$ to denote the model we aim to evaluate, which takes an input of the image and predicts the label.

We use $g(\mathbf{x}, \mathbf{b})$ to denote an image generation system, which takes an input of the starting image $\mathbf{x}$ to generate another image $\widehat{\mathbf{x}}$ within the computation budget $\mathbf{b}$. The generation process is performed as an optimization process to maximize a scoring function $\alpha(\widehat{\mathbf{x}}, \mathbf{z})$ that evaluates the alignment between the generated image and generation goal $\mathbf{z}$ guiding the perturbation process. The higher the score is, the better the alignment is. Thus, the generation process is formalized as

$$\widehat{\mathbf{x}} = \underset{\widehat{\mathbf{x}}=g(\mathbf{x},\mathbf{b}),\mathbf{b}<\mathbf{B}}{\arg\max} \ \alpha(g(\mathbf{x},\mathbf{b}),\mathbf{z}),$$

where $\mathbf{B}$ denotes the allowed computation budget for one sample. This budget will constrain the generated image not far from the starting image so that the generated one does not converge to a trivial solution that maximizes the scoring function.

In addition, we choose the model classification loss $l(\theta(\widehat{\mathbf{x}}), \mathbf{y})$ as $\mathbf{z}$. Therefore, the scoring function essentially maximizes the loss of a given image in the direction of a different class.

Finally, to maintain the unknown image-label structure of the images, we leverage the power of the pretrained giant models to scope the generation process: the generated images must be considered within the same class by the pretrained model, denoted as $h(\widehat{\mathbf{x}})$, which takes in the input of the image and makes a prediction.

---

**Algorithm 1** Perturbed Image Generation with Foundation Models

---

**Input:** $(\mathbf{X}, \mathbf{Y})$, $\theta$, $g$, $h$, total number of iterations $\mathbf{B}$
**Output:** generated dataset $(\widehat{\mathbf{X}}, \mathbf{Y})$
**for** each $(\mathbf{x}, \mathbf{y})$ in $(\mathbf{X}, \mathbf{Y})$ **do**
    generate $\widehat{\mathbf{x}}_0 = g(\mathbf{x}, \mathbf{b}_0; \theta)$
    **if** $h(\widehat{\mathbf{x}}_0) = \mathbf{y}$ **then**
        set $\widehat{\mathbf{x}} = \widehat{\mathbf{x}}_0$
        **for** iteration $\mathbf{b}_t < \mathbf{B}$ **do**
            generate $\widehat{\mathbf{x}}_t = g(\widehat{\mathbf{x}}_{t-1}, \mathbf{b}_t; \theta)$
            **if** $h(\widehat{\mathbf{x}}_t) = \mathbf{y}$ **then**
                set $\widehat{\mathbf{x}} = \widehat{\mathbf{x}}_t$
            **else**
                set $\widehat{\mathbf{x}} = \widehat{\mathbf{x}}_{t-1}$
                exit FOR loop
            **end if**
        **end for**
    **else**
        set $\widehat{\mathbf{x}} = \mathbf{x}$
    **end if**
    use $(\widehat{\mathbf{x}}, \mathbf{y})$ to construct $(\widehat{\mathbf{X}}, \mathbf{Y})$
**end for**

---

Connecting all the components above, the generation process will aim to optimize the following:

$$\widehat{\mathbf{x}} = \underset{\widehat{\mathbf{x}}=g(\mathbf{x},\mathbf{b}),\mathbf{b}<\mathbf{B},\mathbf{z}=l(\theta(\widehat{\mathbf{x}}),\mathbf{y})}{\arg\max} \ \alpha(g(\mathbf{x},\mathbf{b}),\mathbf{z}),$$
$$\text{subject to} \quad h(\widehat{\mathbf{x}}) = \mathbf{y}.$$

Our method is generic and agnostic to the choices of the three major components, namely $\theta$, $g$, and $h$. For example, the $g$ component can vary from something as simple as basic transformations adding noises or rotating images to a sophisticated method to transfer the style of the images; on the other hand, the $h$ component can vary from an approach with high reliability and low efficiency such as actually outsourcing the annotation process to human labors to the other polarity of simply assuming a large-scale pretrained model can function plausibly as a human.

In the next part, we will introduce our concrete choices of $g$ and $h$ leading to the later empirical results, which build upon the recent advances of vision research.

### 3.2 ENGINEERING SPECIFICATION

We use VQGAN (Esser et al., 2021) as the image generation system $g(\mathbf{x}, \mathbf{b})$, and the $g(\mathbf{x}, \mathbf{b})$ is boosted by the evaluated model $\theta(\mathbf{x})$ serving as the $\alpha(\widehat{\mathbf{x}}, \mathbf{z})$ to guide the generation process, where $\mathbf{z} = l(\theta(\widehat{\mathbf{x}}), \mathbf{y})$ is the model classification loss on current perturbed images.

The generation is an iterative process guided by the scoring function: at each iteration, the system adds more style-wise transformations to the result of the previous iteration. Therefore, the total number of iterations allowed is denoted as the budget $\mathbf{B}$ (see Section 4.5 for details of finding the best perturbation). In practice, the value of budget $\mathbf{B}$ is set based on the resource concerns.

To guarantee the image-label structure of images, we consider using foundation models, *e.g.,* the CLIP (Radford et al., 2021) model, to serve as $h$, and design the text fragment input of CLIP to be *"an image of {class}"*. However, given the CLIP model's less-than-perfect zero-shot accuracy on most of the base datasets (Radford et al., 2021), there exists a potential risk of introducing a label noise to the generated test set. Therefore, in order to reduce the dependency of robustness evaluation on the robustness of the specific foundation model, we employ a majority voting mechanism across an ensemble of multiple foundation models to validate the correctness of labels assigned to the generated images. In our experiments, we assemble a zoo of foundation models, including the CLIP model, ConvNeXt-T-CvSt (Singh et al., 2023) from the RobustBench Leaderboard (Croce et al., 2020), and CoCa (Yu et al., 2022) from the robust foundation models leaderboard[1]. The ensemble of these foundation models is drawn from diverse sources and exhibits variability, thus enhancing the credibility of label validation for the generated images through a collective majority vote. Afterwards, we directly optimize VQGAN encoder space which is guided by our scoring function. We show the algorithm in Algorithm 1.

### 3.3 MEASURING ROBUSTNESS

**Foundation Model-oriented Robustness (FMR).** By design, the image-label structures of perturbed images will be maintained by the foundation model. Thus, a smaller accuracy drop on the perturbed images indicates more similar predictions to foundation models. To precisely define FMR, we introduce perturbed accuracy (PA), the accuracy on the perturbed images that our generative model successfully produces. As the standard accuracy on clean test set (SA) may influence PA to some extent, to disentangle PA from SA, we normalize PA with SA as FMR:

$$\text{FMR} = \frac{\text{PA}}{\text{SA}} \times 100\%$$

In settings where the foundation model is human labors, FMR measures the robustness difference between the evaluated model and human perception. In our experiment setting, FMR measures the robustness difference between models trained on fixed datasets (the tested model) and the models trained on unfiltered, highly varied, and highly noisy data (the zoo of foundation models).

At last, we devote a short paragraph to kindly remind some readers that, despite the alluring idea of designing systems that forgo the usages of underlying image-label structure or foundation model, it has been proved or argued multiple times that it is impossible to create that knowledge with nothing but data, in either context of machine learning (Locatello et al., 2019; Mahajan et al., 2019; Wang et al., 2021; Zhao et al., 2022; 2023) or causality (Bareinboim et al., 2020; Xia et al., 2021), (Pearl, 2009, Sec. 1.4).

## 4 EXPERIMENTS - EVALUATION AND UNDERSTANDING OF MODELS

### 4.1 EXPERIMENT SETUP

We consider four different scenarios, ranging from the basic benchmark MNIST (LeCun et al., 1998), through CIFAR10 (Krizhevsky et al., 2009), 9-class ImageNet (Santurkar et al., 2019), to full-fledged 1000-class ImageNet (Deng et al., 2009). For ImageNet, we resize all images to $224 \times 224$ px. We also center and re-scale the color values with $\mu_{RGB} = [0.485, 0.456, 0.406]$ and $\sigma = [0.229, 0.224, 0.225]$. The perturbation step size for each iteration is 0.001. The total number of iterations allowed (computation budget **B**) is 50.

For each of the experiment, we report a set of three results:

- Standard Accuracy (SA): the clean test accuracy of the evaluated model.
- Perturbed Accuracy (PA): accuracy on the images that our generation process successfully produces a perturbed image.
- Foundation Model-oriented Robustness (FMR): robustness of the model compared with the foundation model.

---

[1] https://paperswithcode.com/sota/zero-shot-transfer-image-classification-on-4

## 4.2 ROBUSTNESS EVALUATION FOR STANDARD VISION MODELS

We consider a large range of models (Appedix M) and evaluate pre-trained variants of a LeNet architecture (LeCun et al., 1998) for the MNIST experiment and ResNet architecture (He et al., 2016a) for the remaining experiments. For ImageNet experiment, we also consider pretrained transformer variants of ViT (Dosovitskiy et al., 2020), Swin (Liu et al., 2021), Twins (Chu et al., 2021), Visformer (Chen et al., 2021) and DeiT (Touvron et al., 2021) from the *timm* library (Wightman, 2019). We evaluate the recent ConvNeXt (Liu et al., 2022) as well. All models are trained on the ILSVRC2012 subset of IN comprised of 1.2 million images in the training and a total of 1000 classes (Deng et al., 2009; Russakovsky et al., 2015).

We report our results in Table 1. As expected, these models can barely maintain their performances when tested on data from different distributions, as shown by many previous works (*e.g.,* Geirhos et al., 2019; Hendrycks & Dietterich, 2019).

Interestingly, on ImageNet, though both transformer-variants models and vanilla CNN-architecture model, *i.e.,* ResNet, attain similar clean image accuracy, transformer-variants substantially outperform ResNet50 in terms of FMR under our dynamic evaluation protocol. We conjecture such a performance gap partly originated from the differences in training setups; more specifically, it may be resulted by the

Table 1: The robustness test of standard models. We note 1) there exists a performance gap between standard models and the foundation model, and 2) transformer-variants outperform the vanilla ResNet in terms of FMR.

| Data | Model | SA | PA | FMR |
|------|-------|-----|-----|-----|
| MNIST | LeNet | 99.09 | 24.76 | 24.99 |
| CIFAR10 | ResNet18 | 95.38 | 49.30 | 51.69 |
| 9-class IN | ResNet18 | 92.30 | 24.89 | 26.97 |
| ImageNet | ResNet50 | 76.26 | 30.59 | 40.11 |
| | ViT | 82.40 | 39.57 | 48.02 |
| | DeiT | 78.57 | 41.18 | 52.41 |
| | Twins | 80.53 | 46.47 | 57.71 |
| | Visformer | 79.88 | 45.71 | 57.22 |
| | Swin | 81.67 | 54.93 | 67.26 |
| | ConvNeXt | 82.05 | 45.44 | 55.38 |

fact transformer-variants by default use strong data augmentation strategies while ResNet50 use none of them. The augmentation strategies (*e.g.,* Mixup (Zhang et al., 2017), Cutmix (Yun et al., 2019) and Random Erasing (Zhong et al., 2020), *etc.*) already naively introduce OOD samples during training, therefore are potentially helpful for securing model robustness towards data shifts. When equiping with the similar data augmentation strategies, CNN-architecture model, *i.e.,* ConvNext, has achieved comparable performance in terms of FMR. This hypothesis has also been verified in recent works (Bai et al., 2021; Wang et al., 2022c). We will offer more discussions on the robustness enhancing methods in Section 4.3.

Surprisingly, we notice a large difference within the transformer family in the proposed FMR metric. Despite Swin Transformer's suboptimal accuracy on the standard dataset, it achieves the best FMR. One possible reason for this phenomenon may due to their internal architecture that are related to the self-attention mechanism. Therefore, we conduct in-depth analysis on the effects of head numbers. The results reveal that increased head numbers enhance expressivity and robustness, albeit at the expense of clean accuracy. More details can be found in Appendix C.

Besides comparing performance between different standard models, FMR brings us the chance to directly compare models with the foundation model. Across all of our experiments, the FMR shows the significant gap between models and the foundation model, which is trained on the unfiltered and highly varied data, seemingly suggesting that training with a more diverse dataset would help with robustness. This overarching trend has also been identified in (Taori et al., 2020). However, quantifying when and why training with more data helps is still an interesting open question.

## 4.3 ROBUSTNESS EVALUATION FOR ROBUST VISION MODELS

Recently, some techniques have been introduced to cope with corruptions or style shifts. To investigate whether those OOD robust models can maintain the performance in our dynamic evaluation, we evaluate the pretrained ResNet50 models combining with the five leading methods from the ImageNet-C leaderboard, namely, Stylized ImageNet training (SIN; (Geirhos et al., 2019)), adversarial noise training (ANT; (Rusak et al.)), a combination of ANT and SIN (ANT+SIN; (Rusak et al.)), optimized data augmentation using Augmix (AugMix; (Hendrycks et al., 2019)), DeepAugment (Deep-

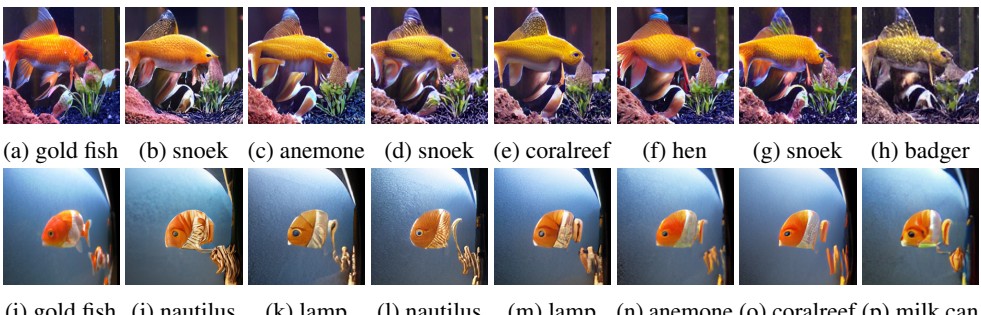

(a) gold fish    (b) snoek    (c) anemone    (d) snoek    (e) coralreef    (f) hen    (g) snoek    (h) badger

(i) gold fish    (j) nautilus    (k) lamp    (l) nautilus    (m) lamp    (n) anemone (o) coralreef (p) milk can

Figure 2: Visualization of the images generated by our system in evaluating the common corruption robust model, with the original image shown (left image of each row). The caption for each image is either the original label or the predicted label by the corresponding model. The evaluated models are SIN, ANT, ANT+SIN, Augmix, DeepAug, DeepAug+AM and DAT from left to right.

Aug; (Hendrycks et al., 2021a)), a combination of Augmix and DeepAug (DeepAug+AM; (Hendrycks et al., 2021a)) and Discrete Adversarial Training (DAT; (Mao et al., 2022b)).

The results are displayed in Table 2. Surprisingly, we find that some common corruption robust models, *i.e.,* SIN, ANT, ANT+SIN, fail to maintain their power under our dynamic evaluation protocol. We take the SIN method as an example. The FMR of SIN method is $40.07$, which is even lower than that of a vanilla ResNet50. These methods are well fitted in the benchmark ImageNet-C, verifying the weakness of relying on fixed benchmarks to rank methods. The selected best method may not be a true reflection of the real world, but a model well fits certain datasets, which in turn proves the necessity of our dynamic evaluation protocol.

Table 2: The robustness test of generated countefactual images for OOD robust models. $SA^*$ represents the model's top-1 accuracy on ImageNet-C dataset. We note that applying DAT yields the best FMR under our dynamic evaluation protocol.

| Model | SA | $SA^*$ | PA | FMR |
|---|---|---|---|---|
| ResNet50 | 76.26 | 39.20 | 30.59 | 40.11 |
| ANT | 76.26 | 50.41 | 30.61 | 40.14 |
| SIN | 76.24 | 45.19 | 30.55 | 40.07 |
| ANT+SIN | 76.26 | 52.60 | 31.09 | 40.77 |
| DeepAug | 76.26 | 52.60 | 33.24 | 43.59 |
| Augmix | 76.73 | 48.31 | 38.89 | 50.68 |
| DeepAug+AM | 76.68 | 58.10 | 42.60 | 55.56 |
| DAT | 76.57 | 68.00 | 52.57 | 68.66 |

DeepAug, Augmix, DeepAug+AM perform better than SIN and ANT methods in terms of FMR as they dynamically perturb the datasets, alleviating the hazards of "model selection with test set" to some extent. DAT outperform others in terms of FMR, which validates the effectiveness of perturbation in the meaningful symbolic space rather than the continuous pixel space. However, their performance is limited by the variations of the perturbations allowed, resulting in marginal improvements compared with ResNet50.

Besides, we visualize the perturbed images generated according to the evaluated style-shift robust models in Figure 2. More results and the discussion on the realism of the generated images are shown in Appendix O and P. We have the following observations:

**Preservation of Local Textual Details.** Recent studies highlight that CNNs often prioritize object textures over shapes for learning (Gatys et al., 2015; Ballester & Araujo, 2016; Gatys et al., 2017; Geirhos et al., 2019; Wang et al., 2020b). Our perturbed images retain misleading textures, making evaluation more challenging as textures become a nuisance rather than predictive. For instance, in Figure 2f, we generated images with skin textures resembling chicken skin, which misleads the ResNet with DeepAug method.

**Generalization to Shape Perturbations.** Our attack dynamically adjusts intensity using the model's gradient, affecting both texture and shape while preserving image-label structures. This results in

Table 3: Study of different image generator choices on ImageNet dataset. The numbers of PA and FMR are reported. The results are consistent under different image generator configurations.

| Model | ADM | | Improved DDPM | | Efficient-VDVAE | | StyleGAN-XL | | VQGAN | |
|---|---|---|---|---|---|---|---|---|---|---|
| | PA | FMR | PA | FMR | PA | FMR | PA | FMR | PA | FMR |
| ResNet50 | 32.36 | 42.43 | 31.43 | 41.21 | 30.28 | 39.71 | 31.65 | 41.50 | 32.09 | 42.08 |
| ANT | 31.88 | 41.80 | 32.54 | 42.67 | 31.29 | 41.03 | 31.94 | 41.88 | 31.65 | 41.50 |
| SIN | 32.17 | 42.20 | 32.05 | 42.04 | 31.15 | 40.86 | 31.39 | 41.17 | 31.64 | 41.50 |
| ANT+SIN | 32.47 | 42.58 | 33.50 | 43.93 | 32.68 | 42.85 | 33.01 | 43.29 | 32.15 | 42.16 |
| DeepAug | 33.32 | 43.69 | 34.39 | 45.10 | 33.83 | 44.36 | 34.46 | 45.19 | 34.30 | 44.98 |
| Augmix | 39.47 | 51.44 | 40.16 | 52.34 | 39.95 | 52.07 | 39.30 | 51.22 | 40.01 | 52.14 |
| DeepAug+AM | 43.43 | 56.64 | 43.17 | 56.30 | 41.32 | 53.89 | 41.71 | 54.39 | 43.65 | 56.92 |

successful model attacks with shape-perturbed images, as demonstrated in the SIN (Figure 2b and Figure 2j) and ANT+SIN (Figure 2d and Figure 2l) examples.

**Recognition of Model Properties.** By integrating various methods, we generate more complex perturbed images, such as those combining DeepAug's chicken-like head with Augmix's skin patterns (Figure 2h), demonstrating our method's ability to adapt to model properties for challenging evaluations. This shows our protocol dynamically tailors attacks to model specifics, producing perturbed images that reveal weaknesses beyond static benchmarks, *i.e.,* ImageNet-C.

## 4.4 UNDERSTANDING THE PROPERTIES OF OUR EVALUATION SYSTEM

We continue to investigate several properties of the models in the next couple sections. To save space, we will mainly present the results on CIFAR10 experiment here and save the details to the appendix:

- In Appendix D, we explore the transferability of the generated images and validate the reliability of the FMR metric. The results of a reasonable transferability suggest that our method of generating images is not model-specificity, and can be potentially used in a broader scope: we can leverage the method to generate a static set of images and set a benchmark to help the development of robustness methods.
- In Appendix F, we compare the vanilla model to a model trained by PGD (Madry et al., 2017). We find that these two models process the data differently. However, their robustness weaknesses are exposed to a similar degree by our test system.
- In Appendix G, we investigate enhancing evaluated robustness by training the model with images generated by our evaluation system. Due to computational constraints, we use a static image set for training, which indeed improves model robustness in our system.
- We also notice that the generated images tend to shift the color of the original images, so we tested the robustness of grayscale models in Appendix H, the results suggest removing the color channel will not improve robustness performances.

## 4.5 EXPERIMENTS REGARDING METHOD CONFIGURATION

**Generator Configuration.** We conduct ablation study on the generator choice to agree on the performance ranking in Table 1 and Table 2. We consider several image generator architechitures, namely, variational autoencoder (VAE) (Kingma & Welling, 2013; Rezende et al., 2014) like Efficient-VDVAE (Hazami et al., 2022), diffusion models (Sohl-Dickstein et al., 2015) like Improved DDPM (Nichol & Dhariwal, 2021) and ADM (Dhariwal & Nichol, 2021), and GAN like StyleGAN-XL (Sauer et al., 2022). As shown in Table 3, we find that the conclusion is consistent under different generator choices, which validates the correctness of our conclusions in Section 4.2 and Section 4.3.

**Sparse VQGAN.** In resource-constrained scenarios, we enhance efficiency by sparsifying VQGAN, discovering that only 0.69% of dimensions significantly impact style. By masking the remaining 99.31%, we create a sparse VQGAN submodel, reducing runtime by 12.7% on 9-class ImageNet and 28.5% on ImageNet, making our protocol viable even with limited computing resources. Further details are in the Appendix I.

**Step Size.** The optimal step size varies with the computation budget (B). Under limited budgets, a larger step size is necessary but may highlight model limitations, affecting evaluation outcomes. With ample B, a smaller step size can alleviate these issues, proving sufficient for practical applications, as detailed in Appendix J.

## 5    DISCUSSION

### 5.1    DISCUSSION ON THE BIAS ISSUES

**Selection Bias.** In previous sections, we have mentioned that ranking models based on the fixed datasets will potentially lead to the selection bias of methods (Duda et al., 1973; Friedman et al., 2001). While our dynamic evaluation protocol help mitigates this issue, it is inevitable to introduce other biases when we select specific generators and foundation models. Here, we provide more analyses and discussions.

**Bias towards Generators.** As our evaluation protocol requires an image generator, the quality or diversity of the generated images may be bounded by the choice of generator. However, Table 3 shows the consistent conclusions made in the paper, which verifies that the proposed method is robust to the choice of generator.

**Bias towards the Foundation Models.** In Appendix J, we take the CLIP model as an example and explore the category unbalance issue. We observe its performance is affected by imbalanced online sample distributions, leading to perturbed images of varied difficulty. Fortunately, our model configurations significantly mitigate this issue (see Appendix J), proving effective for real-world use. Additionally, using an ensemble of foundation models enhances this mitigation.

**Bias of the Metric.** As the generated samples are biased to the zoo of foundation models' zero-shot performance, "PA" and "FMR" scores will also be biased. In Appendix A, we conduct a theoretical analysis to guarantee the correctness of the proposed method. Our theoretical analysis confirms that while both traditional datasets and foundation model zoos can approximate true distributions, the latter achieves this with less variance. Hence, we advocate that bias towards foundation model zoos is preferable to conventional datasets.

**Potential Negative Impacts.** Although the bias incurred by the zoo of foundation models is less detrimental than the biases arisen from fixed benchmark datasets, a more detailed discussion on the potential negative impacts is necessary. Therefore, we discuss the potential negative impacts as well as the societal bias of relying on large models in Appendix K.

### 5.2    DISCUSSION ON THE EFFECTS OF FOUNDATION MODEL'S ZERO-SHOT PERFORMANCE

**Domain Gap Concerns.**  Despite the zero-shot strengths of our foundation model zoo, it may underperform in niche areas, *e.g.,* medical imaging, where general knowledge falls short. However, our framework's adaptable design allows for the easy inclusion of domain-specific pre-trained models, providing a versatile solution for a wide range of applications.

**Zero-shot Adversarial Robustness Concerns.**

Foundation models like CLIP are vulnerable to adversarial attacks (Mao et al., 2022a), potentially undermining evaluation effectiveness if attackers access and manipulate CLIP's weights. In Appendix L, our study into CLIP's zero-shot adversarial robustness reveals it outperforms XCiT-L12 (Debenedetti et al., 2022) in resilience against FGSM attacks, despite susceptibility to classification changes. However, gradient masking and other simple techniques can safeguard CLIP in production, significantly reducing white-box attack risks. For black-box attacks, CLIP demonstrates strong resilience (See Appendix L). By integrating a robust model ensemble and employing a majority vote for image-label validation, our approach enhances security. Therefore, CLIP, particularly when safeguarded by weight and gradient protection techniques and supported by a robust model ensemble, stands as a strong candidate for the ideal foundation model to preserve image-label integrity currently.

## 6    CONCLUSION

In this paper, we first summarized the common practices of model evaluation strategies for robust vision machine learning. We then discussed three desiderata of the robustness evaluation protocol. Further, we offered a simple method that can fulfill these three desiderata at the same time, serving the purpose of evaluating vision models' robustness across generic *i.i.d* benchmarks, without requirement on the prior knowledge of the underlying image-label structure depicted by the images, although relying on a zoo of foundation models.

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

# A   DISTRIBUTION ESTIMATION

## A.1   PROBLEM FORMULATION

Given an unknown ground truth distribution:

$$\mathbf{P} = \text{Unknown}(\mu, \Sigma) \tag{1}$$

where $\mu \in \mathbb{R}^p$ and $\Sigma \in \mathbb{R}^{p \times p}$.

All the samples in our study are sampled from this distribution.

We use $\mathbf{X}_k$ to denote the $k^{\text{th}}$ dataset, with $n_k$ samples, and we use $\mathbf{x}_{k,i}$ to denote the $i^{\text{th}}$ sample in it.

We aim to consider the estimation of $\mu$ from two different models. The conventional smaller model which operates on only one dataset, and WLOG, we assume the smaller model works on $\mathbf{X}_0$; and the bigger, zoo of CLIP-style models, which operates on a collection of datasets, we say it works on $m$ datasets, i.e., $\{\mathbf{X}_1, \mathbf{X}_2, \mathbf{X}_3, \ldots, \mathbf{X}_m\}$, we will compare $\mathbb{E}[\widehat{\mu_0} - \mu]$ and $\mathbb{E}[\widehat{\mu_{\text{CLIP}}} - \mu]$, $\text{VAR}(\widehat{\mu_0})$ and $\text{VAR}(\widehat{\mu_{\text{CLIP}}})$, $\mathbb{E}[\widehat{\Sigma_0} - \Sigma]$ and $\mathbb{E}[\widehat{\Sigma_{\text{CLIP}}} - \Sigma]$ and $\text{VAR}(\widehat{\Sigma_0})$ and $\text{VAR}(\widehat{\Sigma_{\text{CLIP}}})$.

**Assumption I**   Due to dataset collection bias, we assume that, while all the data are sampled with the fixed distribution above, the bias of dataset collection will introduce a bias in the estimation of the true parameter $\mu$, therefore

$$\widehat{\mu_i} = \mu + \epsilon_i \tag{2}$$

where

$$\widehat{\mu_i} := \frac{1}{n_i} \sum_j^{n_i} \mathbf{x}_{i,j} \tag{3}$$

and

$$\epsilon_i \sim N(\mathbf{0}, \mathbf{I}) \tag{4}$$

**Assumption II**   Due to dataset collection bias, we assume that, while all the data are sampled with the fixed distribution above, the bias of dataset collection will introduce a bias in the estimation of the true parameter $\Sigma$, therefore

$$\widehat{\Sigma_i} = \epsilon_i' \Sigma \tag{5}$$

where

$$\widehat{\Sigma_i} := \frac{1}{n_i} \sum_j^{n_i} [(\mathbf{x}_{i,j} - \widehat{\mu_i})^T (\mathbf{x}_{i,j} - \widehat{\mu_i})] \tag{6}$$

and

$$\epsilon_i' \sim \text{Exp}(\mathbf{1}), \tag{7}$$

**Proposition A.1.** *Under Assumptions I and II, we have estimators*

$$\mathbb{E}[\widehat{\mu_{\text{CLIP}}} - \mu] = \mathbb{E}[\widehat{\mu_0} - \mu], \qquad \mathbb{E}[\widehat{\Sigma_{\text{CLIP}}} - \Sigma] = \mathbb{E}[\widehat{\Sigma_0} - \Sigma]$$

$$\text{VAR}(\widehat{\mu_{\text{CLIP}}}) \leq \text{VAR}(\widehat{\mu_0}), \qquad \text{VAR}(\widehat{\Sigma_{\text{CLIP}}}) \leq \text{VAR}(\widehat{\Sigma_0})$$

*where $\leq$ holds element-wise.*

*Proof.* **Estimation of $\mu$.** Under Assumptions I and II, we have

$$\widehat{\mu_0} = \mu + \epsilon_0 \tag{8}$$

We can obtain $\mathbb{E}[\widehat{\mu_0} - \mu]$ and $\text{VAR}(\widehat{\mu_0})$ by marginalizing out the randomness introduced by $\epsilon$:

$$\mathbb{E}[\widehat{\mu_0} - \mu] = \mathbb{E}[\mu + \epsilon_0 - \mu] = \mathbb{E}[\epsilon_0] = \mathbf{0}. \tag{9}$$

$$
\begin{aligned}
\mathrm{VAR}(\widehat{\mu_0}) &= \mathbb{E}[\widehat{\mu_0}^2] - \mathbb{E}^2[\widehat{\mu_0}] \\
&= \mathbb{E}[(\mu + \epsilon_0)^2] - \mathbb{E}^2[(\mu + \epsilon_0)] \\
&= \mathbb{E}[\mu^2 + 2\mu\epsilon_0 + \epsilon_0^2] - (\mu + \mathbb{E}[\epsilon_0])^2 \\
&= \mathbb{E}[\epsilon_0^2] - \mathbb{E}^2[\epsilon_0] \\
&= \mathrm{VAR}(\epsilon_0) \\
&= \mathbf{I}
\end{aligned}
\tag{10}
$$

For $\mathbb{E}[\widehat{\mu_{\mathrm{CLIP}}} - \mu]$ and $\mathrm{VAR}(\widehat{\mu_{\mathrm{CLIP}}})$, we have:

$$
\mathbb{E}[\widehat{\mu_{\mathrm{CLIP}}} - \mu] = \mathbb{E}[\frac{\sum_i^m \epsilon_i n_i}{\sum_i^m n_i}] = \frac{\sum_i^m \mathbb{E}[\epsilon_i] n_i}{\sum_i^m n_i} = \mathbf{0}.
\tag{11}
$$

and

$$
\begin{aligned}
\mathrm{VAR}(\widehat{\mu_{\mathrm{CLIP}}}) &= \mathbb{E}[\widehat{\mu_{\mathrm{CLIP}}}^2] - \mathbb{E}^2[\widehat{\mu_{\mathrm{CLIP}}}] \\
&= \mathbb{E}[(\mu + \epsilon_{\mathrm{CLIP}})^2] - \mathbb{E}^2[(\mu + \epsilon_{\mathrm{CLIP}})] \\
&= \mathbb{E}[\mu^2 + 2\mu\epsilon_{\mathrm{CLIP}} + \epsilon_{\mathrm{CLIP}}^2] - (\mu + \mathbb{E}[\epsilon_{\mathrm{CLIP}}])^2 \\
&= \mathbb{E}[\epsilon_{\mathrm{CLIP}}^2] - \mathbb{E}^2[\epsilon_{\mathrm{CLIP}}]
\end{aligned}
\tag{12}
$$

Since $\mathbb{E}[\epsilon_{\mathrm{CLIP}}] = \mathbb{E}[\widehat{\mu_{\mathrm{CLIP}}} - \mu] = 0$, we have:

$$
\mathrm{VAR}(\widehat{\mu_{\mathrm{CLIP}}}) = \mathbb{E}[\epsilon_{\mathrm{CLIP}}^2] = \mathbb{E}[(\frac{\sum_i^m \epsilon_i n_i}{\sum_i^m n_i})^2]
\tag{13}
$$

When we expand the square of sum, we will get the many squared terms (which are left finally) and many more that involves at least one $\mathbb{E}[\epsilon_i]\mathbf{z}$, where $\mathbf{z}$ can be any arbitrary stuff, and then since $\mathbb{E}[\epsilon_i] = \mathbf{0}$, $\mathbf{z}$ won't matter. Therefore, we have:

$$
\mathrm{VAR}(\widehat{\mu_{\mathrm{CLIP}}}) = \mathbb{E}[(\frac{\sum_i^m \epsilon_i n_i}{\sum_i^m n_i})^2] = \frac{\sum_i^m n_i^2 \mathbb{E}[\epsilon_i^2]}{(\sum_i^m n_i)^2}
\tag{14}
$$

Since $n_i \geq 1$ for $i = 1, 2, ..., m$, we have $\sum_i^m n_i^2 \leq (\sum_i^m n_i)^2$.

Therefore,

$$
\mathrm{VAR}(\widehat{\mu_{\mathrm{CLIP}}}) = \frac{\sum_i^m n_i^2 \mathbb{E}[\epsilon_i^2]}{(\sum_i^m n_i)^2} \leq \frac{(\sum_i^m n_i)^2 \mathbb{E}[\epsilon_i^2]}{(\sum_i^m n_i)^2} = \mathbb{E}[\epsilon_i^2] = \mathbf{I}
\tag{15}
$$

**Estimation of $\Sigma$.** We can obtain $\mathbb{E}[\widehat{\Sigma_0} - \Sigma]$ and $\mathrm{VAR}(\widehat{\Sigma_0})$ by marginalizing out the randomness introduced by $\epsilon'$:

$$
\mathbb{E}[\widehat{\Sigma_0} - \Sigma] = \mathbb{E}[\epsilon_0' \Sigma - \Sigma] = \mathbb{E}[(\epsilon_0' - 1)\Sigma] = \mathbb{E}[\epsilon_0' - 1]\mathbb{E}[\Sigma] = \mathbf{0}.
\tag{16}
$$

$$
\begin{aligned}
\mathrm{VAR}(\widehat{\Sigma_0}) &= \mathbb{E}[\widehat{\Sigma_0}^2] - \mathbb{E}^2[\widehat{\Sigma_0}] \\
&= \mathbb{E}[(\epsilon_0' \Sigma)^2] - \mathbb{E}^2[\epsilon_0' \Sigma] \\
&= \mathbb{E}[\epsilon_0'^2 \Sigma^2] - \mathbb{E}^2[\epsilon_0']\mathbb{E}^2[\Sigma] \\
&= \mathbb{E}[\epsilon_0'^2]\mathbb{E}[\Sigma^2] - \mathbb{E}^2[\epsilon_0']\mathbb{E}^2[\Sigma] \\
&= 2\mathbb{E}[\Sigma^2] - \mathbb{E}^2[\Sigma] \\
&= 2\Sigma^2 - \Sigma^2 \\
&= \Sigma^2
\end{aligned}
\tag{17}
$$

For $\mathbb{E}[\widehat{\Sigma_{\text{CLIP}}} - \Sigma]$ and $\text{VAR}(\widehat{\Sigma_{\text{CLIP}}})$, we have:

$$\mathbb{E}[\widehat{\Sigma_{\text{CLIP}}} - \Sigma] = \mathbb{E}[\epsilon'_{\text{CLIP}}\Sigma - \Sigma] = \mathbb{E}[(\epsilon'_{\text{CLIP}} - 1)\Sigma] = \mathbb{E}[\epsilon'_{\text{CLIP}} - 1]\mathbb{E}[\Sigma] = \mathbf{0}. \tag{18}$$

$$\begin{aligned}
\text{VAR}(\widehat{\Sigma_{\text{CLIP}}}) &= \mathbb{E}[\widehat{\Sigma_{\text{CLIP}}}^2] - \mathbb{E}^2[\widehat{\Sigma_{\text{CLIP}}}] \\
&= \mathbb{E}[(\epsilon'_{\text{CLIP}}\Sigma)^2] - \mathbb{E}^2[\epsilon'_{\text{CLIP}}\Sigma] \\
&= \mathbb{E}[\epsilon'^2_{\text{CLIP}}\Sigma^2] - \mathbb{E}^2[\epsilon'_{\text{CLIP}}]\mathbb{E}^2[\Sigma] \\
&= \mathbb{E}[\epsilon'^2_{\text{CLIP}}]\mathbb{E}[\Sigma^2] - \mathbb{E}^2[\Sigma] \tag{19}
\end{aligned}$$

Consider that

$$\widehat{\Sigma_{\text{CLIP}}} = \frac{\Sigma_i^m \Sigma_j^{n_i}(\mathbf{x}_{i,j} - \widehat{\mu_{\text{CLIP}}})^2}{\Sigma_i^m n_i} = \frac{\Sigma_i^m \epsilon^2_{\text{CLIP}} n_i}{\Sigma_i^m n_i} = \epsilon'_{\text{CLIP}}\Sigma \tag{20}$$

We will have:

$$\epsilon'_{\text{CLIP}} = \frac{\epsilon^2_{\text{CLIP}}}{\Sigma} \tag{21}$$

Thus, we have $\epsilon'^2_{\text{CLIP}} = \frac{\epsilon^4_{\text{CLIP}}}{\Sigma^2}$. Next, we will compute $\mathbb{E}[\epsilon'^2_{\text{CLIP}}]$ as follows:

$$\begin{aligned}
\mathbb{E}[\epsilon'^2_{\text{CLIP}}] &= \mathbb{E}[\frac{\epsilon^4_{\text{CLIP}}}{\Sigma^2}] \\
&= \frac{\mathbb{E}[\epsilon^4_{\text{CLIP}}]}{\Sigma^2} \tag{22}
\end{aligned}$$

By definition, we have:

$$\epsilon_{\text{CLIP}} = \frac{\Sigma_i^m \Sigma_j^{n_i}(\mathbf{x}_{i,j} - \widehat{\mu_i})}{\Sigma_i^m n_i} \tag{23}$$

Therefore,

$$\epsilon^2_{\text{CLIP}} = \frac{(\Sigma_i^m \Sigma_j^{n_i}(\mathbf{x}_{i,j} - \widehat{\mu_i}))^2}{(\Sigma_i^m n_i)^2} \tag{24}$$

As the value of $x_{i,j} - \widehat{\mu_i}$ can be either positive or negative, we have:

$$\epsilon^2_{\text{CLIP}} \leq \frac{\Sigma_i^m \Sigma_j^{n_i}(\mathbf{x}_{i,j} - \widehat{\mu_i})^2}{\Sigma_i^m n_i} \frac{1}{\Sigma_i^m n_i} \tag{25}$$

Since both $(x_{i,j} - \widehat{\mu_i})^2$ and $n_i$ are positive values, we further have:

$$\epsilon^2_{\text{CLIP}} \leq \Sigma_i^m \frac{\Sigma_j^{n_i}(\mathbf{x}_{i,j} - \widehat{\mu_i})^2)}{n_i} \frac{1}{\Sigma_i^m n_i} = \frac{\Sigma_i^m \widehat{\Sigma_i}}{\Sigma_i^m n_i} = \frac{\Sigma_i^m \epsilon'_i \Sigma}{\Sigma_i^m n_i} \tag{26}$$

Thus, we can obtain

$$\epsilon^4_{\text{CLIP}} \leq \frac{(\Sigma_i^m \epsilon'_i \Sigma)^2}{(\Sigma_i^m n_i)^2} = \frac{(\Sigma_i^m \epsilon'_i)^2 \Sigma^2}{(\Sigma_i^m n_i)^2} \tag{27}$$

Therefore, we have:

$$\mathbb{E}[\epsilon^4_{\text{CLIP}}] \leq \mathbb{E}[\frac{(\Sigma_i^m \epsilon'_i)^2 \Sigma^2}{(\Sigma_i^m n_i)^2}] = \frac{\mathbb{E}[(\Sigma_i^m \epsilon'_i)^2]\Sigma^2}{(\Sigma_i^m n_i)^2} \tag{28}$$

By Assumption A.1, $\epsilon'_i \sim \text{Exp}(\mathbf{1})$, we have $\mathbb{E}[\epsilon'_i] = 1$ and $\text{VAR}(\epsilon'_i) = 1$.

Since $\epsilon'_i$ are independent with each other, we have:

$$\begin{aligned}
\mathbb{E}[(\Sigma_i^m \epsilon'_i)^2] &= \text{VAR}(\Sigma_i^m \epsilon'_i) + \mathbb{E}^2[\Sigma_i^m \epsilon'_i] \\
&= \Sigma_i^m \text{VAR}(\epsilon'_i) + (\Sigma_i^m \mathbb{E}[\epsilon'_i])^2 \\
&= m + m^2 \tag{29}
\end{aligned}$$

Substituting Eq. 29 into Eq. 28, we have:

$$\mathbb{E}[\epsilon_{\text{CLIP}}^4] \leq \frac{m + m^2}{(\Sigma_i^m n_i)^2} \Sigma^2 \tag{30}$$

Since $n_i \geq 1$ for $i = 1, 2, ..., m$, we have $\Sigma_i^m n_i \geq m$ and $(\Sigma_i^m n_i)^2 \geq m^2$.

Since $m \geq 1$, we have: $m^2 \geq m$.

Therefore,

$$\mathbb{E}[\epsilon_{\text{CLIP}}^4] \leq \frac{m + m^2}{m^2} \Sigma^2 \leq \frac{2m^2}{m^2} \Sigma^2 = 2\Sigma^2 \tag{31}$$

Substituting Eq. 31 into Eq. 22, we have:

$$\mathbb{E}[\epsilon_{\text{CLIP}}^{'2}] \leq \frac{2\Sigma^2}{\Sigma^2} = 2 \tag{32}$$

Substituting Eq. 32 into Eq. 19, we have:

$$\text{VAR}(\widehat{\Sigma_{\text{CLIP}}}) = \mathbb{E}[\epsilon_{\text{CLIP}}^{'2}]\mathbb{E}[\Sigma^2] - \mathbb{E}^2[\Sigma] \leq 2\mathbb{E}[\Sigma^2] - \mathbb{E}^2[\Sigma] = \mathbb{E}[\Sigma] = \Sigma \tag{33}$$

We summarize the above results as follows: For conventional fixed dataset estimators, we have:

$$\mathbb{E}[\widehat{\mu_0} - \mu] = \mathbf{0}$$
$$\text{VAR}(\widehat{\mu_0}) = \mathbf{I}$$
$$\mathbb{E}[\widehat{\Sigma_0} - \Sigma] = \mathbf{0}$$
$$\text{VAR}(\widehat{\Sigma_0}) = \Sigma^2$$

For CLIP-style estimators, we have:

$$\mathbb{E}[\widehat{\mu_{\text{CLIP}}} - \mu] = \mathbf{0}$$
$$\text{VAR}(\widehat{\mu_{\text{CLIP}}}) \leq \mathbf{I}$$
$$\mathbb{E}[\widehat{\Sigma_{\text{CLIP}}} - \Sigma] = \mathbf{0}$$
$$\text{VAR}(\widehat{\Sigma_{\text{CLIP}}}) \leq \Sigma,$$

where $\leq$ holds element-wise. $\qquad\square$

The results show that, both conventional estimator and zoo of CLIP-style estimator can recover the true $\mu, \Sigma$ of the unknown distribution, but zoo of CLIP-style estimator will have a lower variance, which is more stable to accomplish the task. This conclusion holds for any distributions.

With these theoretical evidence, we kindly argue that biased towards the zoo of CLIP-style models is better than biased on conventional fixed datasets. In addition, recent advances in incorporating the foundation model into various tasks (Liu et al., 2023; Zhang et al., 2023; Bose et al., 2023) also reveals that the community has utilized the foundation model on a large scale and pays little attention on these biases.

## B    NOTES ON THE EXPERIMENTAL SETUP

### B.1    NOTES ON MODELS

Note that we only re-evaluate existing model checkpoints, and hence do not perform any hyperparameter tuning for evaluated models. Our model evaluations are done on 8 NVIDIA V100 GPUs. With our Sparsified VQGAN model, our method is also feasible to work with a small amount of GPU resources. As shown in Appendix I, the proposed protocol can work on a single NVIDIA V100 GPU efficiently.

Our method is generally parameter-free except for the computation budget and perturbation step size. In our experiments, the computation budget is the maximum iteration number of Sparse VQGAN. We consider the predefined value to be 50, as it guarantees the degree of perturbation with acceptable time costs. We provide the experiment for step size configuration in Section 4.5.

## C   In-depth Analysis on the Transformer Family

In Table 1, we notice a large difference between the methods in the proposed FMR metric, even within the transformer family. After checking the distribution of misclassified perturbed images of different models, we find that these images are rather random and do not reveal any obvious "weak classes". One possible reason for this phenomenon may due to their internal architecture that are related to the self-attention (Self-Att) mechanism. Many current Vision Transformer architectures adopt a multi-head self-attention (MHSA) design where each head tends to focus on different object components. In some sense, MHSA can be interpreted as a mixture of information bottlenecks (IB) where the stacked Self-Att modules in Vision Transformers can be broadly regarded as an iterative repeat of the IB process which promotes grouping and noise filtering. More details of the connection between the Self-Att and IB can be found in ((Zhou et al., 2022a), Sec.2.3). As revealed in (Zhou et al., 2022a), having more heads leads to improved expressivity and robustness. But the reduced channel number per head also causes decreased clean accuracy. The best trade-off is achieved with 32 channels per head.

Table 4 illustrates the head number configurations of various models employed in our experiment.

Table 4: Details of head numbers configurations.)

| Model | Head Number |
| --- | --- |
| ViT | 12 |
| DeiT | 12 |
| Twins | (3,6,12,24) |
| Visformer | 6 |
| Swin | (4,8,16,32) |

Swin Transformer exhibits the highest number of heads among them. Despite its suboptimal accuracy on the standard dataset, it achieves the best FMR. This corroborates the finding in (Zhou et al., 2022a) that increased head numbers enhance expressivity and robustness, albeit at the expense of clean accuracy.

To further verify the impact of head numbers, we trained Swin Transformer with varying head configurations and obtained the following results in Table 5.

Table 5: The performance of Swin Transformer with different head number configurations. We find that increased heads enhance expressivity and robustness.)

| #Params | Head Number | SA | FMR |
| --- | --- | --- | --- |
| 88M | (2,4,8,16) | 80.82 | 64.85 |
| 88M | (3,6,12,24) | 81.98 | 67.48 |
| 88M | (4,8,16,32) | 81.67 | 69.73 |
| 88M | (5,10,20,40) | 81.05 | 69.97 |

With comparable numbers of parameters, we observe that their accuracies on the standard dataset are relatively similar. With the augmentation of head numbers, the FMR value also escalates, which validates our hypothesis that increased heads enhance expressivity and robustness.

## D    Transferability of Generated Images

We first study whether our generated images are model-specific, since the generation of the images involves the gradient of the original model. We train several architectures, namely EfficientNet (Tan & Le, 2019), MobileNet (Howard et al., 2017), SimpleDLA (Yu et al., 2018), VGG19 (Simonyan & Zisserman, 2014), PreActResNet (He et al., 2016b), GoogLeNet (Szegedy et al., 2015), and DenseNet121 (Huang et al., 2017) and test these models with the images that generated when testing ResNet. We also train another ResNet following the same procedure to check the transferability across different runs in one architecture.

**Transferability of the generated images.** Table 6 shows a reasonable transferability of the generated images as the FMR are all lower than the SA, although we can also observe an improvement over the FMR when tested in the new models. These results suggest that our method of generating images can be potentially used in a broader scope: we can also leverage the method to generate a static set of images and set a benchmark dataset to help the development of robustness methods.

Table 6: Performances of transferability.

| Model | SA | PA | FMR |
|---|---|---|---|
| ResNet | 95.38 | 51.67 | 54.17 |
| ResNet | 94.67 | 56.09 | 59.25 |
| DenseNet | 94.26 | 60.48 | 64.17 |
| SimpleDLA | 92.25 | 61.03 | 66.16 |
| GoogLeNet | 92.06 | 61.10 | 66.38 |
| PreActResNet | 90.91 | 61.14 | 67.25 |
| EfficientNet | 91.37 | 62.57 | 68.48 |
| MobileNet | 91.63 | 62.97 | 68.72 |
| VGG | 93.54 | 66.01 | 70.57 |

**Reliability of the FMR metric.** Moreover, these results contribute to the validation of the reliability of the FMR metric: given that each model's FMR gets computed using a different test set, it is not clear why FMR would be a reliable metric that can be used to compare two models. In this experiment, however, the models are tested using the same fixed test set that was initially generated during the evaluation on ResNet. Remarkably, the strong correlation observed between FMR and PA at the fixed test sets lends credence to the reliability of the FMR metric.

**New findings.** In addition, our results might potentially help mitigate a debate on whether more accurate architectures are naturally more robust: on one hand, we have results showing that more accurate architectures indeed lead to better empirical performances on certain (usually fixed) robustness benchmarks (Rozsa et al., 2016; Hendrycks & Dietterich, 2019); while on the other hand, some counterpoints suggest the higher robustness numerical performances are only because these models capture more non-robust features that also happen exist in the fixed benchmarks (Tsipras et al., 2018; Wang et al., 2020b; Taori et al., 2020). Table 6 show some examples to support the latter argument: in particular, we notice that VGG, while ranked in the middle of the accuracy ladder, interestingly stands out when tested with generated images. These results continue to support our argument that a dynamic robustness test scenario can help reveal more properties of the model.

## E    Initiating with Adversarial Attacked Images

Since our method using the gradient of the evaluated model reminds readers about the gradient-based attack methods in adversarial robustness literature, we test whether initiating the perturbation process with an adversarial example will further degrade the accuracy.

Table 7: Results on whether initiating with adversarial images ($\epsilon = 0.003$).

| Data | SA | FMR |
|---|---|---|
| regular | 95.38 | 57.80 |
| w. FGSM | 95.30 | 65.79 |

We first generate the images with FGSM attack (Goodfellow et al., 2015). Table 7. shows that initiating with the FGSM adversarial examples barely affect the FMR, which is probably because the major style-wise perturbation will erase the imperceptible perturbations the adversarial examples introduce.

## F    Adversarially Robust Models

With evidence suggesting the adversarially robust models are considered more human perceptually aligned (Engstrom et al., 2019; Zhang & Zhu, 2019; Wang et al., 2020b), we compare the vanilla model to a model trained by PGD (Madry et al., 2017) ($\ell_\infty$ norm smaller than 0.03).

As shown in Table 8, adversarially trained model and vanillaly trained model indeed process the data differently: the transferability of the generated images between these two regimes can barely hold. In particular, the PGD model can almost maintain its performances when tested with the images generated by the vanilla model.

However, despite the differences, the PGD model's robustness weak spots are exposed to a similar degree with the vanilla model by our test system: the FMR of the vanilla

Table 8: Performances comparison with vanilla model and PGD trained model.

| Data | Model | SA | FMR |
|------|-------|-------|-------|
| Van. | Van. | 95.38 | 57.79 |
| | PGD | 85.70 | 95.96 |
| PGD | Van. | 95.38 | 62.41 |
| | PGD | 85.70 | 66.18 |

model and the PGD model are only 57.79 and 66.18, respectively. We believe this result can further help advocate our belief that the robustness test needs to be a dynamic process generating images conditioning on the model to test, and thus further help validate the importance of our contribution.

## G  AUGMENTATION THROUGH STATIC ADVERSARIAL TRAINING

Intuitively, inspired by the success of adversarial training (Madry et al., 2017) in defending models against adversarial attacks, a natural method to improve the empirical performances under our new test protocol is to augment the training data with perturbed training images generated by the same process. We aim to validate the effectiveness of this method here.

However, the computational load of generation process is not ideal to serve the standard adversarial training strategy, and we can only have one copy of the perturbed training samples. Fortunately, we notice that some recent advances in training with data augmentation can help learn robust representations with a limited scope of augmented samples (Wang et al., 2020a), which we use here.

We report our results in Table 9. The first thing we observe is that the model trained with the augmentation data offered through our approach could preserve a relatively higher performance (FMR 89.13) when testing with the perturbed images generated according to the vanilla model. Since we have shown the perturbed samples have a reasonable transferability in the main manuscript, this result indicates the robustness we brought when training with the perturbed images generated by our approach.

Table 9: Test performances of the model trained in a vanilla manner (denoted as Van.) or with augmentation data offered through our approach (marked by the second column). We report two sets of performances, split by whether the perturbed images are generated according to the vanilla model or the augmented model (marked by the first column).

| Data | Model | SA | FMR |
|------|-------|-------|-------|
| Van. | Van. | 95.38 | 57.82 |
| | Aug | 87.41 | 89.13 |
| Aug. | Van. | 95.38 | 58.03 |
| | Aug | 87.41 | 78.61 |

In addition, when tested with the perturbed images generated according to the augmented model, the augmented model displays a marked resilience (FMR 78.61) in the face of these perturbations compared with the model trained in a vanilla manner (FMR 58.03). Nevertheless, it is noteworthy that the augmented model's performance does exhibit a discernible decline under these circumstances, which once more underscores the efficacy of our approach.

## H  GRAYSCALE MODELS

Our previous visualization suggests that a shortcut the perturbed generation system can take is to significantly shift the color of the images, for which a grey-scale model should easily maintain the performance. Thus, we train a grayscale model by changing the ResNet input channel to be 1 and transforming the input images to be grayscale upon feeding into the model. We report the results in Table 10.

Interestingly, we notice that the grayscale model cannot defend against the shift introduced by our system by ignoring the color information. On the contrary, it seems to encourage our system to generate more perturbed images that can lower the performances.

In addition, we visualize some perturbed images generated according to each model and show them in Figure 3. We can see some evidence that the grayscale model forces the generation system to

Table 10: Test performances of the model trained in a vanilla manner (denoted as Van.) or with grayscale model. We report two sets of performances, split by whether the perturbed images are generated according to the vanilla model or the grayscale one (marked by the first column).

| Data | Model | SA | FMR |
|------|-------|-------|-------|
| Van. | Van. | 95.38 | 57.79 |
|      | Gray | 93.52 | 66.06 |
| Gray | Van. | 95.38 | 67.48 |
|      | Gray | 93.52 | 44.76 |

focus more on the shape of the object and less of the color of the images. We find it particularly interesting that our system sometimes generates different images differently for different models while the resulting images deceive the respective model to make the same prediction.

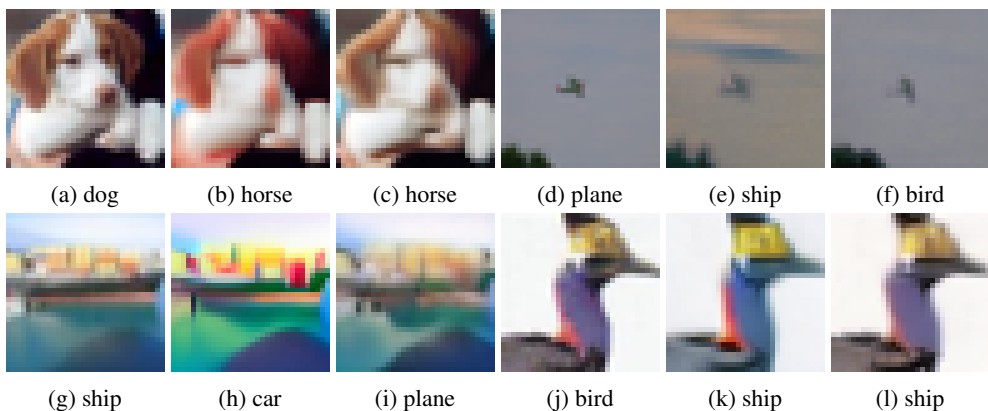

| (a) dog | (b) horse | (c) horse | (d) plane | (e) ship | (f) bird |

| (g) ship | (h) car | (i) plane | (j) bird | (k) ship | (l) ship |

Figure 3: Visualization of the perturbed images generated by our system in evaluating the vanilla model (middle image of each group) and the grayscale model (third image of each group), with the original image shown. The caption for each image is either the original label or the predicted label by the corresponding model.

## I  SPARSE SUBMODEL OF VQGAN FOR EFFICIENT PERTURBATION

While our method will function properly as described above, we notice that the generation process still has a potential limitation: the bound-free perturbation of VQGAN will sometimes perturb the semantics of the images, generating results that will be rejected by the foundation model later and thus leading to a waste of computational efforts.

To counter this challenge, we use a sparse variable selection method to analyze the embedding dimensions of VQGAN to identify a subset of dimensions that is mainly responsible for the non-semantic variations.

In particular, with a dataset $(\mathbf{X}, \mathbf{Y})$ of $n$ samples, we first use VQGAN to generate a style-transferred dataset $(\mathbf{X}', \mathbf{Y})$. During the generation process, we preserve the latent representations of input samples after the VQGAN encoder in the original dataset. We also preserve the final latent representations before the VQGAN decoder that are quantized after the iterations in the style-transferred dataset. Then, we create a new dataset $(\mathbb{E}, \mathbf{L})$ of $2n$ samples, for each sample $(\mathbf{e}, l) \in (\mathbb{E}, \mathbf{L})$, $\mathbf{e}$ is the latent representation for the sample (from either the original dataset or the style-transferred one), and $l$ is labelled as 0 if the sample is from the original dataset and 1 if the style-transferred dataset.

Then, we train $\ell_1$ regularized logistic regression model to classify the samples of $(\mathbb{E}, \mathbf{L})$. With $\mathbf{w}$ denoting the weights of the model, we solve the following problem

$$\arg\min_{\mathbf{w}} \sum_{(\mathbf{e}, l) \in (\mathbb{E}, \mathbf{L})} l(\mathbf{e}\mathbf{w}, l) + \lambda \|\mathbf{w}\|_1,$$

Table 11: Classification results between vanilla and perturbed images with LASSO.

| Data | Sparsity | Test score |
|---|---|---|
| MNIST | 97.99 | 78.50 |
| CIFAR-10 | 98.45 | 78.00 |
| 9-class ImageNet | 99.31 | 72.00 |
| ImageNet | 99.32 | 69.00 |

and the sparse pattern (zeros or not) of $\mathbf{w}$ will inform us about which dimensions are for the style.

We generate the flattened latent representations of input images after the VQGAN Encoder with negative labels. Following Algorithm 1, we generate the flattened final latent representations before the VQGAN decoder with positive labels. Altogether, we form a binary classification dataset where the number of positive and negative samples is balanced. The positive samples are the latent representations of perturbed images while the negative samples are the latent representations of input images. We set the split ratio of train and test set to be $0.8 : 0.2$. We perform the explorations on various datasets, i.e. MNIST, CIFAR-10, 9-class ImageNet and ImageNet.

The classification model we consider is LASSO[2] as it enables automatically feature selection with strong interpretability. We set the regularization strength to be 36.36. We adopt saga (Defazio et al., 2014) as the solver to use in the optimization process. The classification results are shown in Table 11.

We observe that the coefficient matrix of features can be far sparser than we expect. We take the result of 9-class ImageNet as an example. Surprisingly, we find that almost 99.31% dimensions in average could be discarded when making judgements. We argue the preserved 0.69% dimensions are highly correlated to VQGAN perturbation. Therefore, we keep the corresponding 99.31% dimensions unchanged and only let the rest 0.69% dimensions participate in computation. Our computation loads could be significantly reduced while still maintain the competitive performance compared with the unmasked version[3].

We conduct the run-time experiments on a single NVIDIA V100 GPU. Following our experiment setting, we evaluate a vanilla ResNet-18 on 9-class ImageNet and a vanilla ResNet-50 on ImageNet. As shown in Table 12, the run-time on ImageNet can be reduced by 28.5% with our sparse VQGAN. Compared with large-scale masked dimensions (*i.e.,* 99.31%), we attribute the relatively incremental run-time improvement (*i.e.,* 12.7% on 9-class ImageNet, 28.5% on ImageNet) to the fact that we have to perform mask and unmask operations each time when calculating the model gradient, which offsets the calculation efficiency brought by the sparse VQGAN to a certain extent.

Table 12: Run-time Comparision between VQGAN and Sparse VQGAN.

| Method | Time | |
|---|---|---|
| | 9-class ImageNet | ImageNet |
| VQGAN | $521.5 \pm 1.2$s | $52602.4 \pm 2.7$s |
| Sparse VQGAN | $455.4 \pm 1.2$s | $40946.1 \pm 2.7$s |
| *Improv.* | 12.7% | 28.5% |

## J    ANALYSIS OF SAMPLES THAT ARE MISCLASSIFIED BY THE MODEL

We notice that, the CLIP model has been influenced by the imbalance sample distributions across the Internet.

In this experiment, we choose a larger step size so that the foundation model may not be able to maintain the image-label structure at the first perturbation. We report the Validation Rate (VR) which

---

[2]Although LASSO is originally a regression model, we probabilize the regression values to get the final classification results.

[3]We note that the overlapping degree of the preserved dimensions for each dataset is not high, which means that we need to specify these dimensions when facing new datasets.

is the percentage of images validated by the foundation model that maintains the image-label structure. (In our official configurations, the step size value is small enough that the VR on each dataset is always 1. Therefore, we omit this value in the main experiments.) We present the results on 9-class ImageNet experiment to show the details for each category.

Table 13: Details of test on 9-class ImageNet for vanilla ResNet-18 (step size is 0.1, computation budget $\mathbf{B}$ is 50)

| Type | SA | VR | FMR |
|---|---|---|---|
| Dog | 93.33 | 95.33 | 17.98 |
| Cat | 96.67 | 94.00 | 31.55 |
| Frog | 85.33 | 80.67 | 20.34 |
| Turtle | 84.67 | 78.67 | 29.03 |
| Bird | 91.33 | 96.00 | 28.13 |
| Primate | 96.00 | 48.00 | 62.21 |
| Fish | 94.00 | 76.67 | 45.33 |
| Crab | 96.00 | 87.33 | 19.87 |
| Insect | 93.33 | 78.00 | 33.88 |
| Total | 92.30 | 81.63 | 30.28 |

Table 13 shows that the VR values for most categories are still higher than 80%, some even reach 95%, which means we produce sufficient number of perturbed images. However, we notice that the VR value for *primate* images is quite lower compared with other categories, indicating around 52% perturbed *primate* images are blocked by the orcle.

As shown in Table 13, the FMR value for each category significantly drops compared with the SA value, indicating the weakness of trained models. An interesting finding is that the FMR value for *Primate* images are quite higher than other categories, given the fact that more perturbed *Primate* images are blocked by the foundation model. We attribute it to the limitation of foundation models. As the CLIP model has been influenced by the imbalance sample distributions across the Internet, it could only handle easy perturbed samples well. Therefore, the perturbed images preserved would be those that can be easily classified by the models.

Table 14: Details of test on 9-class ImageNet for vanilla ResNet-18 (step size is 0.001, computation budget $\mathbf{B}$ is 50)

| Type | SA | VR | FMR |
|---|---|---|---|
| Dog | 93.33 | 100.00 | 18.09 |
| Cat | 96.67 | 100.00 | 28.60 |
| Frog | 85.33 | 100.00 | 20.72 |
| Turtle | 84.67 | 100.00 | 24.80 |
| Bird | 91.33 | 100.00 | 27.68 |
| Primate | 96.00 | 100.00 | 27.11 |
| Fish | 94.00 | 100.00 | 25.13 |
| Crab | 96.00 | 100.00 | 19.15 |
| Insect | 93.33 | 100.00 | 23.16 |
| Total | 92.30 | 100.00 | 23.94 |

In our official configuration, we set a relatively smaller step size to perturb the image and obtain enough more perturbed images. As shown in Table 14, using a smaller step size value and enough computation budget barely affect the overall results. In addition, with smller step size, we manage to perturb the image little by little and can get enough more perturbed images (**VR becomes 100 on every category**, indicating that all the images are perturbed and maintained their image-label structure). Admittedly, the foundation model's bias still exists here, e.g., the *Primate* images (FMR = 28.11) are still easier than *Dog* images (FMR = 18.09). However, considering the huge performance gap between the foundation model and the evaluated models, images that are easy for the foundation model are hard enough for the evaluated models (The FMR of *Dogs* and *Primate* images are closer and

smaller compared with those in Table 13), which is sufficiently efficacious for real-world applications. Additionally, the employment of an ensemble of multiple foundation models in our methodology serves to provide a further layer of alleviation for the aforementioned issue.

## K  DISCUSSIONS ON THE SOCIETAL BIAS OF RELYING ON LARGE MODELS

### K.1  POTENTIAL NEGATIVE IMPACTS OF FOUNDATION MODELS

Although the bias incurred by foundation models is less detrimental than the biases arisen from fixed benchmark datasets, a more detailed discussion on the potential negative impacts is necessary. One potential bias of making vision models behave more like the foundation models is that the vision model may inherit the limitations and assumptions of foundation models' training data and objective function. For example, foundation models' training data may not cover all possible visual concepts or scenarios that are relevant to a given task; foundation models' objective function may not align with the desired outcome or evaluation metric of a given task; foundation models' natural language supervision may introduce ambiguities or inconsistencies that affect the model's performance or interpretation. These limitations and assumptions may affect the generalization and robustness of vision models that rely on foundation models. Moreover, we add recent works that especially investigate the bias of foundation models, and guide the readers to it for further warning, e.g., (Menon et al., 2022) and (Zhou et al., 2022b).

### K.2  SOCIETAL BIAS OF RELYING ON LARGE MODELS

Moreover, our method relies on large models, where their societal bias is still unclear, therefore a related discussion would be beneficial.

Large-scale models could leverage the rich knowledge and generalization ability encoded in the training stage. However, one potential societal bias of relying on large models' supervision on preserving the perturbed image could be that it would privilege certain groups or perspectives over others based on social or cultural norms. As the data used to train the pre-trained models may be imbalanced, incomplete, or inaccurate, leading to biased representations of certain groups or concepts, the perturbed images preserved by the pre-trained models may reflect stereotypes, or discrimination against certain groups of people based on their race, gender, age, religion, etc., which may be harmful, offensive, or deceptive to the users. Bridging the gap between the pre-trained model and the evaluated vision models will make the vision models inherit the limitations of pre-trained models, which have adverse consequences for people who are affected by them, such as reinforcing stereotypes, discrimination, or exclusion.

We add recent works that investigate the societal bias of large models, and guide the readers to it for further warning, *e.g.,* (Wang et al., 2022a).

## L  EXPERIMENTS ON THE ZERO-SHOT ADVERSARIAL ROBUSTNESS OF CLIP

We conduct the following experiment to compare the adversarial vulnerability between CLIP and robust ViT-like model pre-trained checkpoints of XCiT-L12 (Debenedetti et al., 2022) from the RobustBench Leaderboard (Croce et al., 2020). The results are shown in Table 15. We find that the vanilla CLIP shows a better robustness performance under our quick experiments through FGSM attack. However, if we continue the attack process, we will eventually obtain the adversary that changes the CLIP's classification decision to the targeted class.

Fortunately, in production, one can use simpler techniques such as gradient masking to protect CLIP's weights from malicious users, thus, the opportunities of the CLIP being attacked from a white-box manner are quite low. In terms of black-box attacks, CLIP actually shows a strong resilience toward the adversarial samples generated for other models, for which we also have some supporting evidence: In Appendix E, we generate the images with the FGSM attack by the tested model. Table 7 shows that initiating with the FGSM adversarial examples barely affects the FMR, which implies that CLIP succeeds in defending these black-box adversarial images and preserving the hard ones such that the FMR does not change significantly (Otherwise, CLIP will discard heavily perturbed images and preserve easy ones with minor perturbation, leading to high FMR values). Furthermore, our approach

Table 15: Comparison of the zero-shot adversarial robustness of CLIP with pretrained robust model. We find that CLIP shows a better robustness performance compared with XCiT-L12. We note that the CLIP's classification decision can be changed to the targeted class as attack continues.

| Step | Target loss | | p[true=0] | | p[target=1] | |
|------|-------------|----------|-----------|----------|-------------|----------|
|      | CLIP | XCiT-L12 | CLIP | XCiT-L12 | CLIP | XCiT-L12 |
| 0 | 8.621 | 4.712 | 0.6749 | 0.7437 | 0.0052 | 0.0728 |
| 20 | 2.715 | 1.605 | 0.5083 | 0.4074 | 0.0986 | 0.2009 |
| 40 | 2.316 | 0.8877 | 0.4007 | 0.2562 | 0.1357 | 0.3116 |
| 60 | 1.684 | 0.7420 | 0.2177 | 0.1407 | 0.2144 | 0.4760 |
| 80 | 1.540 | 0.6520 | 0.1813 | 0.1338 | 0.3335 | 0.5210 |

incorporates an ensemble of foundation models, including robust models such as ConvNext-T-CvSt from the RobustBench Leaderboard, and employs a majority vote mechanism to validate the fidelity of the image-label relationships.

Thus, CLIP, especially when equipped with techniques to protect its weights and gradients, and coupled with an ensemble of robust foundation models, might be the closest one to serve as the ideal foundation models to maintain the image-label structure at this moment.

# M    LIST OF EVALUATED MODELS

The following lists contains all models we evaluated on various datasets with references and links to the corresponding source code.

## M.1    PRETRAINED VQGAN MODEL

We use the checkpoint of vqgan_imagenet_f16_16384 from `https://heibox.uni-heidelberg.de/d/a7530b09fed84f80a887/`

## M.2    PRETRAINED FOUNDATION MODELS

1. Model weights of ViT-B/32 and usage code are taken from `https://github.com/openai/CLIP`

2. CoCa (Yu et al., 2022) `https://github.com/lucidrains/CoCa-pytorch`

3. ConvNeXt-T-CvSt (Singh et al., 2023) `https://github.com/nmndeep/revisiting-at`

## M.3    TIMM MODELS TRAINED ON IMAGENET (WIGHTMAN, 2019)

Weights are taken from `https://github.com/rwightman/pytorch-image-models/tree/master/timm/models`

1. ResNet50 (He et al., 2016a)

2. ViT (Dosovitskiy et al., 2020)

3. DeiT (Touvron et al., 2021)

4. Twins (Chu et al., 2021)

5. Visformer (Chen et al., 2021)

6. Swin (Liu et al., 2021)

7. ConvNeXt (Liu et al., 2022)

### M.4 ROBUST RESNET50 MODELS

1. ResNet50 SIN+IN (Geirhos et al., 2019) `https://github.com/rgeirhos/texture-vs-shape`
2. ResNet50 ANT (Rusak et al.) `https://github.com/bethgelab/game-of-noise`
3. ResNet50 ANT+SIN (Rusak et al.) `https://github.com/bethgelab/game-of-noise`
4. ResNet50 Augmix (Hendrycks et al., 2019) `https://github.com/google-research/augmix`
5. ResNet50 DeepAugment (Hendrycks et al., 2021a) `https://github.com/hendrycks/imagenet-r`
6. ResNet50 DeepAugment+Augmix (Hendrycks et al., 2021a) `https://github.com/hendrycks/imagenet-r`
7. ResNet50 Discrete Adversarial Training (DAT) (Mao et al., 2022b) `https://github.com/alibaba/easyrobust`

### M.5 ADDITIONAL IMAGE GENERATORS

1. Efficient-VDVAE (Hazami et al., 2022) `https://github.com/Rayhane-mamah/Efficient-VDVAE`
2. Improved DDPM (Nichol & Dhariwal, 2021) `https://github.com/open-mmlab/mmgeneration/tree/master/configs/improved_ddpm`
3. ADM (Dhariwal & Nichol, 2021) `https://github.com/openai/guided-diffusion`
4. StyleGAN (Sauer et al., 2022) `https://github.com/autonomousvision/stylegan_xl`

### M.6 PRETRAINED XCIT-L12 MODEL

Model weights of XCiT-L12 (Debenedetti et al., 2022) are taken from `https://github.com/dedeswim/vits-robustness-torch`

## N LEADERBOARDS FOR ROBUST IMAGE MODEL

We launch leaderboards for robust image models. The goal of these leaderboards are as follows:

- To keep on track of state-of-the-art on each adversarial vision task and new model architectures with our dynamic evaluation process.
- To see the comparison of robust vision models at a glance (*e.g.,* performance, speed, size, *etc.*).
- To access their research papers and implementations on different frameworks.

We offer a sample of the robust ImageNet classification leaderboard in supplementary materials.

## O ADDITIONAL PERTURBED IMAGE SAMPLES

In Figure 4, we provide additional perturbed images generated according to each model. We have similar observations to Section 4.3. First, the generated perturbed images exhibit diversity that many other superficial factors of the data would be covered, *i.e.,* texture, shape and styles. Second, our method could recognize the model properties, and automatically generate those hard perturbed images to complete the evaluation.

In addition, the generated images show a reasonable transferability in Table 6, indicating tha our method can be potentially used in a broader scope: we can also leverage the method to generate a static set of images and set a benchmark dataset to help the development of robustness methods.

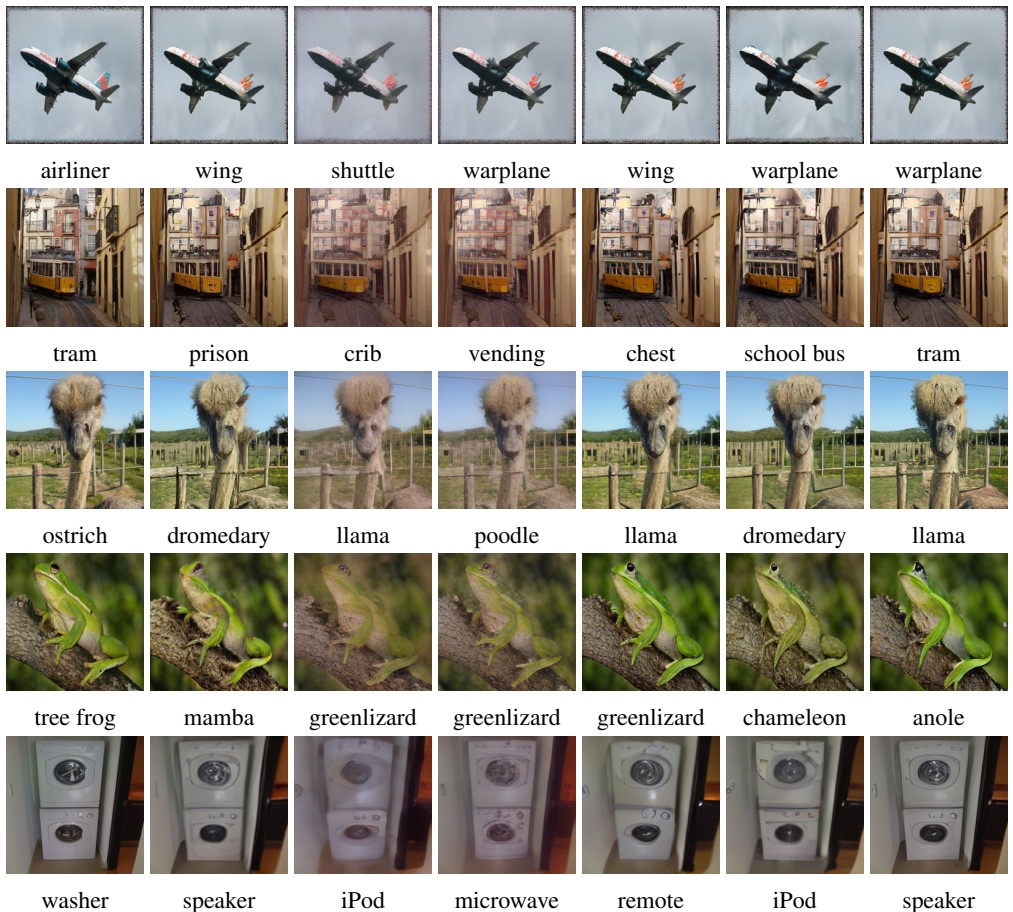

Figure 4: Visualization of the images generated by our system in evaluating the common corruption robust model, with the original image shown (left image of each row). The caption for each image is either the original label or the predicted label by the corresponding model. The evaluated models are SIN, ANT, ANT+SIN, Augmix, DeepAug and DeepAug+AM from left to right.

Therefore, we also offer two static benchmarks in supplementary materials that are generated based on CNN architecture, *i.e.,* ConvNext and transformer variant, *i.e.,* ViT, respectively.

## P    DISCUSSION ON THE REALISM OF THE GENERATED IMAGES

We notice that some generated images look unnatural, as the generated images being realistic is not part of the optimization function. We acknowledge that making the generated images appear more natural will be a further desideratum, as this contributes to enhancing the human-perceptible interpretability.

Nonetheless, the current research agenda of the robustness evaluation community is still to encourage the evaluation to expose the model's weakness, such as to expose and eliminate the model's learning of spurious correlation in rare cases.

Similar evidence can be found in (Xiao et al., 2023), where the authors utilize masked images as counterfactual samples for robust fine-tuning. In this paper, the authors argue that masked images can break the spurious correlation between features and labels that may degrade OOD robustness, and that feature-based distillation with the pre-trained model on these counterfactual samples can achieve a better trade-off between IID and OOD performance. According to our second desideratum, our generated counterfactual images might also look unnatural. However, although it appears unnatural, it is beneficial in uncovering and eliminating spurious correlations for enhancing the model robustness.

