# OpenReview forum: "Foundation Model-oriented Robustness: Robust Image Model Evaluation with Pretrained Models"
_ICLR.cc/2024/Conference — ICLR 2024 poster_

### Official Review · Reviewer_FCU2 · 2023-10-30

**Soundness:** 2 fair
**Presentation:** 3 good
**Contribution:** 2 fair
**Rating:** 6
**Confidence:** 4

**Summary:**

This paper proposed foundation model-oriented robustness, where foundation models are used as the oracle when testing vision models’ robustness. A pipeline is developed to maximize model’s misalignment while preserving semantics with an ensemble of foundation models, and a limiting computation budget. Lastly the authors presented a systematic study of Foundation Model-oriented Robustness (FMR) on ImageNet and ImageNet-C.

**Strengths:**

1. The proposed pipeline made use of existing foundation models as the oracle, addressing several issues of existing testing benchmarks, such as limited to specific curated perturbations or a distribution different from the training set.
2. The flexibility of the proposed method allows robustness evaluation on a wide range of image datasets and vision models.
3. The authors presented a thorough study of vision models (CNN or transformer based), and different augmentation methods.

**Weaknesses:**

1. The perturbations produced by the model seem limited to those barely visible to human beings. Is this desirable? For now all perturbations seem weak, they are small perturbations around the image, barely visible to humans. There are many strong generation models, such as edge-conditioned ControlNet, or inpating based diffusion models that can generate very diverse outputs in terms of colors, and styles. I think these strong perturbations would be very interesting and naturally suitable in this framework.
2. There’s limited understanding of what the perturbed images are. This is important to support the “diversity” argument in the introduction. How similar are the images to ImageNet-C perturbations?
3. It’s challenging to read the texts in Figure 1…

**Questions:**

1. As discussed in the introduction, two main limitations of existing evaluations are the limited type of perturbations, and testing data in a different distribution from training set. How well did this work improve these issues over previous evaluations? Are the perturbations more diverse? For the data distribution, there is also a gap between ImageNet training and foundation model training left unaddressed.

---

> ### Author Response · Authors · 2023-11-16
> **Response 1/2**
>
> We thank the reviewer for the comments and the thorough review. We are grateful that they recognize the flexibility of our method. We are encouraged that they find that our validation is sufficient. Finally, we thank the reviewer for their positive comments and the insightful feedback.
>
> **1. The perturbations produced by the model seem limited to those barely visible to human beings. Is this desirable? For now all perturbations seem weak, they are small perturbations around the image, barely visible to humans. There are many strong generation models, such as edge-conditioned ControlNet, or inpating based diffusion models that can generate very diverse outputs in terms of colors, and styles. I think these strong perturbations would be very interesting and naturally suitable in this framework.**
>
> Thank you for your insightful comments regarding the nature and intensity of the perturbations generated by our model.
>
>
> Please note that the design rationale of our method is that we search for the perturbations with meaningful changes around the image that lead the models to mis-classify. Once these models misclassify the images to a certain amount of perturbation, it will be very likely the models will misclassify with more dramatic changes. Thus, we believe smaller changes are more precise reflections of the evaluated performances.
>
>
> We agree that stronger perturbations would also be interesting. In our experiments, we can control the strength of perturbations by changing the perturbation step size. For example, in Appendix J, we explore the generation of strong perturbations using a larger step size. However, we observe that images with strong perturbations are blocked due to the limitations of foundation models, resulting in the ​​imbalance in the sample distribution. Consequently, the perturbed images preserved would be those that can be easily classified by the models, leading to the unreliablility of evaluation results.
>
>
> Therefore, while we acknowledge the benefits of strong perturbations, our official configurations could maintain a balanced sample distribution, avoid blocking issues, and ensure that the perturbations remain challenging for the evaluated models.
>
> **2. There’s limited understanding of what the perturbed images are. This is important to support the “diversity” argument in the introduction. How similar are the images to ImageNet-C perturbations?**
>
> Thank you for your comment regarding the nature of the perturbed images in our methodology and their relation to the diversity argument presented in the introduction.
>
> ImageNet-C uses a standard set of perturbations like noise, blur, and color variations, applied uniformly across all models. In contrast, our method generates unique perturbations for each model and for each image, often reaching complexities beyond predefined perturbation of images. For example, our approach might create subtle texture changes (Figure 2f) or shape-specific distortions (Figure 2b, 2d, 2l, 2j) that ImageNet-C’s standard perturbations cannot capture.
>
> **3. It’s challenging to read the texts in Figure 1…**
>
> We appreciate this feedback, which helps us improve the overall quality and accessibility of our work. We have revised Image 1 to enhance the clarity of the text.

---

> ### Author Response · Authors · 2023-11-16
> **Response 2/2**
>
> **4. As discussed in the introduction, two main limitations of existing evaluations are the limited type of perturbations, and testing data in a different distribution from training set. How well did this work improve these issues over previous evaluations? Are the perturbations more diverse? For the data distribution, there is also a gap between ImageNet training and foundation model training left unaddressed.**
>
> Thank you for your thoughtful remarks highlighting the crucial aspects of our work in addressing the limitations of existing evaluations.
>
> **"How well did this work improve these issues over previous evaluations"**
>
> 1. **Increased Diversity in Perturbations**: Compared to the limited range of perturbations in datasets like ImageNet-C, our method generated a significantly broader array of perturbed images. For example, our approach might create subtle texture changes (Figure 2f) or shapecontext-specific distortions (Figure 2b, 2d, 2l, 2j) that ImageNet-C’s standard perturbations cannot capture.
>
> 2. **Effectiveness in Exposing Model Weaknesses**: Our method creates personalized perturbed images that are effective in exposing model weaknesses that previous evaluations fail to reveal. For example, in Table 2, robust models that perform well on static ImageNet-C falter under our dynamic evaluation protocol: their FMR value is even lower than that of a vanilla ResNet50.
>
> In addition, these robust methods are well fitted in the benchmark ImageNet-C, verifying the weakness of relying on fixed benchmarks to rank methods. The selected best method may not be a true reflection of the real world, but a model well fits certain datasets, which in turn proves the necessity of our dynamic evaluation protocol.
>
> **"a gap between ImageNet training and foundation model training left unaddressed"**
>
> Our method, especially the theoretical dicussions in appendix, builds upon the assumption that the ImageNet dataset is a subset of the dataset that used to train the foundation models. Our work employs three foundation models: ConvNeXt-T-CvSt, CLIP, and CoCa. ImageNet is indeed a subset of the dataset used for pre-training ConvNeXt-T-CvSt [1]. This direct inclusion of ImageNet in the training dataset ensures there is no significant distribution gap when evaluating performance on ImageNet tasks [2].
>
> [1] Revisiting Adversarial Training for ImageNet: Architectures, Training and Generalization across Threat Models. arXiv:2303.01870. 2023
>
> [2] What makes ImageNet good for transfer learning? arXiv:1608.08614. 2016

---

> ### Author Response · Authors · 2023-11-22
>
> Dear reviewer FCU2,
>
> We are eager to receive your feedback and any suggestions you might have. Your insights would be invaluable in enhancing the quality of this work. If there is any additional information or clarification needed from our end, please feel free to let us know.
>
> Thank you once again for your time and consideration. We look forward to hearing from you soon.
>
> Sincerely,
>
> Authors

---

> ### Comment · Reviewer_FCU2 · 2023-11-22
> **Official comment from Reviewer FCU2**
>
> Thank the authors for the feedback.
>
> I've went through the reviews and discussions from other reviewers and I am convinced that your work proposed a new and interesting way to evaluate model robustness.
>
> I have some follow-up concerns:
>
> 1. **Fairness of comparisons.** As models are evaluated on different images, is this a fair comparison? What happens if we evaluate models on perturbed images from other models? Are images with smaller adversarial perturbations in general easier to defend for this model and other models too?
> 2. **Interpretations from the evaluations.** Are models easier to be adversarially attacked necessarily "less robust"? In general I think robustness are often considered under fixed and known perturbations (e.g., ImageNet-C) or from different domains (e.g., ImageNet-R). When the robustness evaluations is combined with adversarial attacks, it's less clear what's the source of the performance degradation or what's the main factor leading to the wrong predictions. Also could the authors discuss more specifically the implications of the evaluation results on real world problems? More specifically, comparing to ImageNet-C etc.?

---

> ### Author Response · Authors · 2023-11-22
>
> Thank you for your positive feedback and acknowledgment of our work. We are pleased to hear that you find our approach to evaluating model robustness both new and interesting.
>
> For the follow-up concerns:
>
> **1. Fairness of comparisons.**
>
> **"As models are evaluated on different images, is this a fair comparison?"**
>
> Thank you for raising this concern, we believe this is a fair comparison in evaluating how models are deviating from the performance of a shared and fixed foundation model (thus we name our metric “foundation model oriented robustenss”).
>
> **"What happens if we evaluate models on perturbed images from other models? Are images with smaller adversarial perturbations in general easier to defend for this model and other models too?"**
>
> In Appendix D, we specifically address this concern by evaluating models on perturbed images from ResNet. This experiment aims to assess whether the perturbed images are specific to a particular model or exhibit transferability across different models.
>
> The results in Table 6 indicate a reasonable transferability of the generated images as the FMR are all lower than the SA. This result suggests that the perturbed images retain their effectiveness to some extent, even when applied to models other than the one they were specifically designed to deceive.
>
> Crucially, we also observe an improvement over the FMR when these perturbed images are tested in other models. This improvement indicates that these images, including smaller perturbations, are indeed easier for other models to defend against, as they are not specifically designed to target those models.
>
> **2. Interpretations from the evaluations.**
>
> **"Are models easier to be adversarially attacked necessarily "less robust"?"**
>
> Thanks for the question, as the reviewer suggested, the definition of “robustness” has multiple meanings when evaluated with ImageNet-C or with ImageNet-R. Our evaluation metrics opens a new dimension of robustness evaluation. Thus, it’s hard to say one model “less robust” if a model shows inferior performance on one of these evaluation metrics, but not necessarily others.
>
> **"it's less clear what's the source of the performance degradation or what's the main factor leading to the wrong predictions. Also could the authors discuss more specifically the implications of the evaluation results on real world problems? More specifically, comparing to ImageNet-C etc. ?"**
>
> We agree that the evaluation is less interpretable than the evaluation result that ImageNet-C gives, our evaluation metric is designed to have a more dynamic evaluation process that involves multiple factors including the foundational model. However, we do not necessarily consider this as an disadvantage in comparison to ImageNet-C. We consider this as a trade-off: while ImageNet-C give more interpretable evaluation results due to its clear design, it also limits the power of evaluation to be only with the perturbation, it’s limited in further exposing the model’s weakness beyond its design. On the other hand, our evaluation metric can better expose the model’s weakness, as we discussed in Section 4.3, but the interpretation needs to be done case-by-case. We need to visualize the generated sample to understand the behavior of the model.
>
> For example, In Figure 2f, we produce a skin texture similar to chicken skin, and ResNet with DeepAug method is misled by this corruption.
> From a practical perspective for real application, we belive our result suggests that DeepAug (if applied in practice) can further be improved by incorporating more texture-wise perturbations.

---

> ### Comment · Reviewer_FCU2 · 2023-11-23
> **Official comment from Reviewer FCU2**
>
> Thanks to the authors for the feedback. I don't have any further questions.

---

> > ### Author Response · Authors · 2023-11-23
> >
> > Thank you for letting us know we have addressed most of your concerns. If there are any remaining concerns or questions, please do not hesitate to share them with us, and we will happily address them. Otherwise, we kindly request that you consider increasing the score, as the discussion stage is approaching its end.

---

### Official Review · Reviewer_eMG1 · 2023-10-31

**Soundness:** 3 good
**Presentation:** 3 good
**Contribution:** 3 good
**Rating:** 8
**Confidence:** 4

**Summary:**

This paper introduces a novel approach to evaluate machine learning models' robustness. Rather than relying solely on fixed benchmarks, the authors propose a method that directly measures image classification models' performance in comparison to a surrogate oracle, a set of foundational models. They extend image datasets with perturbed samples that maintain the original image-label structure but are distinct from the original set. This approach allows for a more comprehensive evaluation of models' robustness, overcoming limitations associated with fixed benchmarks and constrained perturbations. The paper not only presents evaluation results but also provides insights into model behaviors and the new evaluation strategies employed.

**Strengths:**

The paper discusses the evaluation of model performance, which is an important topic.

**Weaknesses:**

1. The text in Image 1 is too small to discern clearly.
2. I am skeptical about the significance of the paper. I really don't understand why the performance needs to be compared with the foundation model. If there are already foundation model for the given task, is it really necessary to train a new model with similar architecture from scratch?
3. If users need to train different generative models for various downstream tasks, I think this evaluation approach is not user-friendly.
4. Is there a possibility that because the evaluation method proposed in this paper uses VQGAN, and DAT also utilizes VQGAN during training, DAT performs best under this specific evaluation scenario? If that's the case, the evaluation method proposed in this paper might not provide enough insight to the readers.

**Questions:**

See above.

---

> ### Author Response · Authors · 2023-11-16
>
> We thank the reviewer for their comments. We are happy that the reviewer appreciate the novelty of our work. We are encouraged that they recognize the importance of our work. Finally, we thank them for their valuable suggestions.
>
> **1. The text in Image 1 is too small to discern clearly.**
>
> We appreciate this feedback, which helps us improve the overall quality and accessibility of our work. We have revised Image 1 to enhance the clarity of the text.
>
> **2. I am skeptical about the significance of the paper. I really don't understand why the performance needs to be compared with the foundation model. If there are already foundation model for the given task, is it really necessary to train a new model with similar architecture from scratch?**
>
> Thank you for your skepticism regarding the significance of our paper.
>
> We hope to remind the reviewer that, while foundation models represent a significant achievement in domains with abundant resources, they are not universally available or applicable across all domains. In many fields, especially those with less resource availability, creating such large-scale models may not be feasible. However, these fields also need machine learning models that are robust enough for their own applications.
>
> Our evaluation protocol is not necessarily there to bring in the next smaller model that can match the performances of foundation model on natural images, but to encourage the development of such techniques and methods that can get models there, so that these techniques can be used in other domains with less resources.
>
> Last, we believe that even in domains with existing foundation models, new models can offer improvements in terms of efficiency, adaptability, and performance on specific tasks or under unique constraints
> We hope this response adequately addresses your concerns. If you have further skepticism, please let us know.
>
> **3. If users need to train different generative models for various downstream tasks, I think this evaluation approach is not user-friendly.**
>
> Thank you for your feedback regarding the user-friendliness of our evaluation approach.
>
> Contrary to the concern raised, our evaluation approach is designed for flexibility with a 'plug-and-play' feature for generative models. Users do not need to train different generative models for each downstream task. Further, in Table 3, we investigated the performance of several image generator configurations. The results demonstrate consistent performance across different generative models, showing the consistency of our proposed method to the choice of generator.
>
> **4. Is there a possibility that because the evaluation method proposed in this paper uses VQGAN, and DAT also utilizes VQGAN during training, DAT performs best under this specific evaluation scenario? If that's the case, the evaluation method proposed in this paper might not provide enough insight to the readers.**
>
> Thank you for raising this concern about the potential impact of using VQGAN in our evaluation method and its correlation with the performance of DAT.
>
> To address this concern, we evaluate DAT with different generators following the setting in Table 3.
>
> | Model | &nbsp;&nbsp;&nbsp;&nbsp;&nbsp;&nbsp;&nbsp;&nbsp;&nbsp;&nbsp;&nbsp;&nbsp;ADM | &nbsp;Improved DDPM | &nbsp;&nbsp;Efficient-VDVAE |  &nbsp;&nbsp;&nbsp;StyleGAN-XL |  &nbsp;&nbsp;&nbsp;&nbsp;&nbsp;&nbsp;&nbsp;&nbsp;&nbsp;VQGAN |
> | ---- | ---- | ---- | ---- | ---- | ---- |
> |       | PA  &nbsp;&nbsp;&nbsp;&nbsp;&nbsp;&nbsp;&nbsp;&nbsp;&nbsp;&nbsp;&nbsp;&nbsp;&nbsp;&nbsp;&nbsp;&nbsp;&nbsp;&nbsp;&nbsp;&nbsp;FMR | PA &nbsp;&nbsp;&nbsp;&nbsp;&nbsp;&nbsp;&nbsp;&nbsp;&nbsp;&nbsp;&nbsp;&nbsp;&nbsp;&nbsp;&nbsp;&nbsp;&nbsp;&nbsp;&nbsp;&nbsp;FMR | PA &nbsp;&nbsp;&nbsp;&nbsp;&nbsp;&nbsp;&nbsp;&nbsp;&nbsp;&nbsp;&nbsp;&nbsp;&nbsp;&nbsp;&nbsp;&nbsp;&nbsp;&nbsp;&nbsp;&nbsp;FMR | PA &nbsp;&nbsp;&nbsp;&nbsp;&nbsp;&nbsp;&nbsp;&nbsp;&nbsp;&nbsp;&nbsp;&nbsp;&nbsp;&nbsp;&nbsp;&nbsp;&nbsp;&nbsp;&nbsp;&nbsp;FMR | PA &nbsp;&nbsp;&nbsp;&nbsp;&nbsp;&nbsp;&nbsp;&nbsp;&nbsp;&nbsp;&nbsp;&nbsp;&nbsp;&nbsp;&nbsp;&nbsp;&nbsp;&nbsp;&nbsp;&nbsp;&nbsp;FMR |
> | DAT | 53.18 &nbsp;&nbsp;&nbsp;&nbsp;&nbsp;&nbsp;&nbsp;&nbsp;&nbsp;&nbsp;&nbsp;&nbsp; 69.45| 52.91 &nbsp;&nbsp;&nbsp;&nbsp;&nbsp;&nbsp;&nbsp;&nbsp;&nbsp;&nbsp;&nbsp;&nbsp; 69.10 | 52.87 &nbsp;&nbsp;&nbsp;&nbsp;&nbsp;&nbsp;&nbsp;&nbsp;&nbsp;&nbsp;&nbsp;&nbsp; 69.05 | 52.43 &nbsp;&nbsp;&nbsp;&nbsp;&nbsp;&nbsp;&nbsp;&nbsp;&nbsp;&nbsp;&nbsp;&nbsp; 68.47 | 53.03 &nbsp;&nbsp;&nbsp;&nbsp;&nbsp;&nbsp;&nbsp;&nbsp;&nbsp;&nbsp;&nbsp;&nbsp; 69.26|
>
> The results show that DAT consistently outperforms other baseline methods across different generator choices. This consistent performance across various generators suggests that DAT's effectiveness is not because of shared use of VQGAN in its training and our evaluation method.

---

> > ### Comment · Reviewer_eMG1 · 2023-11-21
> >
> > Thanks for the authors' responses, which have addressed every one of my concerns. Therefore, I recommend accepting this manuscript and I have raised the score to 8.

---

> > > ### Author Response · Authors · 2023-11-22
> > >
> > > Dear reviewer eMG1,
> > >
> > > Thanks so much for your positive feedback. We are pleased to hear that our responses have addressed your concerns. Your support and recommendation for accepting our manuscript are greatly appreciated.
> > >
> > > Sincerely,
> > >
> > > Authors

---

### Official Review · Reviewer_LU8n · 2023-11-01

**Soundness:** 3 good
**Presentation:** 3 good
**Contribution:** 3 good
**Rating:** 6
**Confidence:** 3

**Summary:**

The paper proposes a method to generate perturbed images which have to take the "to-be" evaluated models into account and uses the prediction of the pre-trained large foundation models as "oracle/ground-truth/constraints". These perturbed images could serve as "dynamic" evaluation benchmarks for robustness evaluation compared to previously fixed benchmarks.

**Strengths:**

1) The paper is well-organized and the writing is clear and easy to follow;
2) The problem is clearly stated and the experiments have included enough details.
3) The authors have provided a nice discussion, which really benefits readers to better understand this proposed method.

**Weaknesses:**

1) \theta is an input for Algorithm-1, but do not see where it has been used in Algorithm-1.

2) It is not clear to me why this method has any advantages over other benchmarking datasets. The big difference is this proposed method will take the "to-be" evaluated model into account when generating the "adversarial examples". It looks like this proposed method could generate "personalized" "adversarial" examples for the "to-be" evaluated models. So each "to-be" evaluated model may be evaluated on totally different datasets. But in order to fairly compare across different "to-be" evaluated models, the authors normalize their performance by the pre-trained large foundation model? However, I do not see any benefits as to why this evaluation pipeline could better reflect real-world scenarios.

3) It is not clear what "SA" means. Is SA the clean test accuracy of the foundation model? Also, there is an abusive use of "SA" to present two different terms in this paper---- SA for "standard accuracy" and SA for "self-attention".

4) If the perturbed generated images are the images that cause the maximum classification loss of the "to-be" evaluated model, then it will definitely result in a very bad classification performance of the "to-be" evaluated model on the perturbed images. It is not clear to me how the authors do this kind of evaluation. From my current understanding, the generated images have already had the knowledge of the "to-be" evaluated model. Then I doubt how "PA/SA" could be a fair comparison for different "to-be" evaluated models.

5) The overall idea looks very similar to "adversarial training". The big difference is instead of using "norm-bouned" perturbation, here the authors use the prediction of large foundation models as a constraint to limit the perturbation to not go too far. So from this perspective, I only see limited novelty of this proposed method. And even, the perturbed images may be eventually biased towards the foundation models.

6) Could we also report the performance of the "to-be" evaluated models on some of fixed benchmarks as a robustness reference?

**Questions:**

see [weaknesses].

---

> ### Author Response · Authors · 2023-11-16
> **Response 1/3**
>
> We thank the reviewer for the positive comments and the thorough review. We are happy that they appreciate the problem we study, the quality of our work, and the systematic evaluation. We also thank them for their insightful comments and feedback.
>
> **1. \theta is an input for Algorithm-1, but do not see where it has been used in Algorithm-1.**
>
> Thanks for noticing this. We have revised the Algorithm-1 to explicitly state how θ is used in the algorithm to ensure clarity.
>
> As a quick summary, in Algorithm-1, θ represents the model being evaluated. The generation of perturbed images is driven by the goal of maximizing the classification loss of θ. This approach ensures that the generated images are tailored to challenge the specific model θ, thereby allowing us to assess the model's robustness and performance under conditions specifically designed to test its limits.
>
>
>
> **2. It is not clear to me why this method has any advantages over other benchmarking datasets. The big difference is this proposed method will take the "to-be" evaluated model into account when generating the "adversarial examples". It looks like this proposed method could generate "personalized" "adversarial" examples for the "to-be" evaluated models. So each "to-be" evaluated model may be evaluated on totally different datasets. But in order to fairly compare across different "to-be" evaluated models, the authors normalize their performance by the pre-trained large foundation model? However, I do not see any benefits as to why this evaluation pipeline could better reflect real-world scenarios.**
>
> Thank you for your insightful question.
>
> To clarify, our goal with this method is not to claim a complete reflection of the real-world but rather to move closer to the real world than the static benchmark dataset by addressing some of the limitations inherent in static benchmark datasets.
>
> We believe this evaluation will bring in new benefits on several aspects, such as, as the reviewer mentioned, “personalized” “adversarial examples”. And further, we normalize the performances against foundation models for fair comparison across models.
>
> We believe these benefits of our evaluation can be seen from our results.
> For example, As shown in Figure 2, our method creates adversarial examples that are specifically designed to target the weaknesses of each evaluated model. These personalized challenges is effective in exposing model weaknesses that previous evaluations fail to reveal. For example, in Table 2, robust models that perform well on static ImageNet-C falter under our dynamic evaluation protocol: their FMR value is even lower than that of a vanilla ResNet50.
>
> In addition, these methods are well fitted in the benchmark ImageNet-C, verifying the weakness of relying on fixed benchmarks to rank methods. The selected best method may not be a true reflection of the real world, but a model well fits certain datasets, which in turn proves the necessity of our dynamic evaluation protocol.
>
>
> **3. It is not clear what "SA" means. Is SA the clean test accuracy of the foundation model? Also, there is an abusive use of "SA" to present two different terms in this paper---- SA for "standard accuracy" and SA for "self-attention".**
>
> Thank you for your observation about the dual use of "SA" in our paper. To clarify, "SA" in the main paper body consistently stands for "Standard Accuracy" in our context, referring to the clean test accuracy of the evaluated model.
>
> There is another SA that stands for “self attention” in the appendix. We have revised the term "Self-Attention" to "Self-Att" throughout the paper.

---

> ### Author Response · Authors · 2023-11-16
> **Response 2/3**
>
> **4. If the perturbed generated images are the images that cause the maximum classification loss of the "to-be" evaluated model, then it will definitely result in a very bad classification performance of the "to-be" evaluated model on the perturbed images. It is not clear to me how the authors do this kind of evaluation. From my current understanding, the generated images have already had the knowledge of the "to-be" evaluated model. Then I doubt how "PA/SA" could be a fair comparison for different "to-be" evaluated models.**
>
> Thank you for your comments regarding the use of perturbed images and the fairness of the PA/SA comparison in our evaluation methodology.
>
>  **“then it will definitely result in a very bad classification performance of the "to-be" evaluated model on the perturbed images”**
>
> Please notice that this is not necessarily true. When we perturb images, these images are constrained by foundation models to ensure they remain within the same class. Therefore, on one hand, these images do not necessarily lead the “to-be” evaluated model to “very bad classification”. Also, testing against these images (that are “adversarial” for “to-be” models and considered consistent by foundation models) reveals how the evaluated model's performance deviates from foundation models, pinpointing its specific weaknesses.
>
> **“how "PA/SA" could be a fair comparison for different "to-be" evaluated models”**
>
> Please note that, due to the reasons listed above, the PA/SA metric measures how the model deviates from the foundation model. The foundation model serves as an anchor, and how far each model is away from the fixed foundation model is a fair comparison.
>
> **5. The overall idea looks very similar to "adversarial training". The big difference is instead of using "norm-bouned" perturbation, here the authors use the prediction of large foundation models as a constraint to limit the perturbation to not go too far. So from this perspective, I only see limited novelty of this proposed method. And even, the perturbed images may be eventually biased towards the foundation models.**
>
> Thank you for your feedback. We respectfully remind the reviewer that our work is the first to incorporate a generative model and a pre-trained model to generate images for robustness testing.
>
> Furthermore, we would like to emphasize that our work introduces several concrete novelties that we believe are significant:
> Firstly, even under the scope of “adversarial training”, we would like to remind the reviewer that our work is the first to incorporate a generative model and a pre-trained model to for robustness testing under this “adversarial training” concept. We believe this is a significant stride forward from the “norm-bounded” adversarial attacks.
>
> Secondly, to consider the scenarios with limited computing resources, we introduce methods to sparsify the VQGAN and speed up the generation process. Our analysis has enabled us to identify a subset of dimensions within VQGAN that are primarily responsible for non-semantic variations. To our knowledge, this is also a unique improvement of VQGAN for our scenarios.
>
> Thirdly, by leveraging our unique evaluation metric and protocol, we have conducted the first systematic study on the robustness evaluation of deep learning models. This has provided us with valuable insights into the behavior of these models and has opened up new avenues for future research.
>
> Lastly, with our work, we are the first to discuss the desiderata of a robustness evaluation protocol, which serves to mitigate the potential hazards of "model selection with test set." (second paragraph of the background section).
>
> **“the perturbed images may be eventually biased towards the foundation models.”**
>
> in Section 5.1, we have discussed the potential bias towards foundation models: we find that CLIP has been influenced by the imbalance sample distributions across the Internet, resulting in perturbed images with varying degrees of difficulty.
> Fortunately, our model configurations are devised to limit the extent to which perturbations are influenced by the characteristics of the foundation models, as elucidated in Appendix J. Additionally, the employment of an ensemble of multiple foundation models in our methodology provides a further layer of alleviation for this issue.
>
> We believe that these contributions are novel and significant for advancing the field of computer vision and improving the robustness of vision models.

---

> ### Author Response · Authors · 2023-11-16
> **Response 3/3**
>
> **6. Could we also report the performance of the "to-be" evaluated models on some of fixed benchmarks as a robustness reference?**
>
> We appreciate this constructive idea.
>
> To provide the requested data, we refer to existing published works where these models have been evaluated. Specifically,
> * ANT, SIN and ANT+SIN: Performance in terms of mean Corruption Error (mCE) is detailed in Table 9 of [1].
> * AugMix, DeepAugment, and DeepAugment + AugMix: Evaluations can be found in Table 7 of [2].
> * DAT: Performance is available in [3].
>
> In addition, we also report the Top-1 accuracy (SA*) of these models on ImageNet-C  in Table 2, serving as a robust reference point for comparison.
>
> These results are merged into the following table:
>
> | Model | mCE↓ | SA* | SA | PA | FMR |
> | ---- | ---- | ---- | ---- | ---- | ---- |
> | ResNet50 | 77 | 39.20 | 76.26 | 30.59 | 40.11 |
> | ANT | 62 | 50.41 | 76.26 | 30.61 | 40.14 |
> | SIN | 69 | 45.19 | 76.24 | 30.55 | 40.07 |
> | ANT+SIN | 61 | 52.60 | 76.26 | 31.09 | 40.77 |
> | DeepAug | 60.4 | 52.60 | 76.26 | 33.24 | 43.59 |
> | Augmix | 65.3 | 48.31 | 76.73 | 38.89 | 50.68 |
> | DeepAug+AM | 53.6 | 58.10 | 76.68 | 42.60 | 55.56 |
> | DAT | 45.59 | 68.00 | 76.57 | 52.57 | 68.66 |
>
> [1] INCREASING THE ROBUSTNESS OF DNNS AGAINST IMAGE CORRUPTIONS BY PLAYING THE GAME OF NOISE. ICLR 2020
>
> [2] The Many Faces of Robustness: A Critical Analysis of Out-of-Distribution Generalization. ICCV 2021.
>
> [3] EasyRobust: A Comprehensive and Easy-to-use Toolkit for Robust Computer Vision. https://github.com/alibaba/easyrobust 2022

---

> ### Author Response · Authors · 2023-11-22
>
> Dear reviewer LU8n,
>
> We are eager to receive your feedback and any suggestions you might have. Your insights would be invaluable in enhancing the quality of this work. If there is any additional information or clarification needed from our end, please feel free to let us know.
>
> Thank you once again for your time and consideration. We look forward to hearing from you soon.
>
> Sincerely,
>
> Authors

---

> ### Author Response · Authors · 2023-11-23
>
> Dear reviewer LU8n,
>
> As today is the final day of our discussion, we anticipate the opportunity to engage with you. Please let us know if our response has successfully addressed all of your concerns. If so, we would be deeply grateful if you consider raising your score. If not, we will happily answer any additional questions you may have. Thank you!
>
> Sincerely,
>
> Authors

---

### Meta-Review · Area_Chair_k5j2 · 2023-12-11

**Metareview:**

This paper presents studies the problem of generating perturbed images with considering the "to-be" evaluated models, using predictions from pre-trained large foundation models as constraints. It details a technique for generating label-specific stochastic prompts and aligning visual patches with textual prompts through a conditional transport framework.

The paper received positive ratings (8, 6, 6), and all reviewers acknowledged the paper clear organization and writing and comprehensive experiments to show its effectiveness in addressing overfitting in vision-language prompt learning. The concerns raised were also properly addressed in the rebuttal and discussion. Therefore, the reviewers and AC recommended it for acceptance. Congrats to the authors on the nice work!

**Justification For Why Not Higher Score:**

This paper is pretty nice with comprehensive experiments and a clear presentation. While I initially place it in the poster category, I am willing to consider a slightly higher score (Accept spotlight), if the SAC feel it more appropriate.

**Justification For Why Not Lower Score:**

All the reviewers agree this submission is above the acceptance bar.

---

### Decision · Program_Chairs · 2024-01-16

Accept (poster)